# A neural mechanism for detecting object motion during self-motion

**HyungGoo R Kim[1,2,3], Dora E Angelaki[4], Gregory C DeAngelis[2]\***

[1]Department of Biomedical Engineering, Sungkyunkwan University, Suwon, Republic of Korea; [2]Department of Brain and Cognitive Sciences, Center for Visual Science, University of Rochester, Rochester, United States; [3]Center for Neuroscience Imaging Research, Institute for Basic Science, Suwon, Republic of Korea; [4]Center for Neural Science, New York University, New York, United States

**Abstract** Detection of objects that move in a scene is a fundamental computation performed by the visual system. This computation is greatly complicated by observer motion, which causes most objects to move across the retinal image. How the visual system detects scene-relative object motion during self-motion is poorly understood. Human behavioral studies suggest that the visual system may identify local conflicts between motion parallax and binocular disparity cues to depth and may use these signals to detect moving objects. We describe a novel mechanism for performing this computation based on neurons in macaque middle temporal (MT) area with incongruent depth tuning for binocular disparity and motion parallax cues. Neurons with incongruent tuning respond selectively to scene-relative object motion, and their responses are predictive of perceptual decisions when animals are trained to detect a moving object during self-motion. This finding establishes a novel functional role for neurons with incongruent tuning for multiple depth cues.

## Editor's evaluation

This paper will be of broad interest to readers in the field of visual processing. The authors use concurrent psychophysics and single unit recordings, along with modeling, to investigate how visual signals in primate cortical area MT can distinguish between visual motion induced by self-motion and the motion of other objects in the world. The experiments provide an explanation for otherwise puzzling discrepancies in the depth tuning of MT cells.

**\*For correspondence:**
gdeangelis@ur.rochester.edu

**Competing interest:** The authors declare that no competing interests exist.

## Introduction

When an observer moves through the environment, image motion on the retina generally includes components caused by self-motion and objects that move relative to the scene, both of which depend on the depth structure of the scene. Because self-motion typically causes a complex pattern of image motion across the visual field (optic flow, *Gibson et al., 1959*; *Koenderink and van Doorn, 1987*), detecting the movement of objects relative to the world can be a difficult task for the brain to solve. An object that is moving in the world might appear to move faster or slower in the image than objects that are stationary in the scene, depending on the specific viewing geometry. Thus, a critical computational challenge for detecting scene-relative object motion is to identify components of image motion that are not caused by one's self-motion and the static depth structure of the scene. This is a form of causal inference problem (*Shams and Beierholm, 2010*; *French and DeAngelis, 2020*).

Object movement may be relatively easy to distinguish from self-motion when the object's temporal motion profile is clearly different from that of image motion resulting from self-motion (*Layton and Fajen, 2016a*) or when the object moves in a direction that is incompatible with self-motion (*Royden*

*and Connors, 2010*). Neural mechanisms with center-surround interactions in velocity space have been proposed as potential solutions to the problem of detecting object motion under these types of conditions (*Royden and Holloway, 2014*; *Royden et al., 2015*). However, the brain has a remarkable ability to detect object motion even under conditions in which the image velocity of a moving object is very similar to that of stationary background elements during self-motion. *Rushton et al., 2007* demonstrated that object movement relative to the scene 'pops out' when 3D structure is specified by binocular disparity cues but not in the absence of disparity cues. They suggested that disparity cues help the visual system to discount the global flow field resulting from self-motion, thereby identifying object motion. How the brain might achieve this computation has remained a mystery.

We previously reported that many neurons in area MT have incongruent tuning for depth defined by binocular disparity and motion parallax cues (*Nadler et al., 2013*). We speculated that such neurons might play a role in detecting object motion during self-motion by responding selectively to local conflicts between disparity and motion parallax cues (*Nadler et al., 2013*; *Kim et al., 2016a*). Here, we test this hypothesis directly by recording from MT neurons while monkeys perform a task that requires detecting object motion during self-motion. We show that monkeys perform this task based mainly on local differences in depth as cued by disparity and motion parallax. We demonstrate that MT neurons with incongruent tuning for depth based on disparity and motion parallax are generally more sensitive to scene-relative object motion than neurons with congruent tuning. We show that MT neurons that respond with a consistent preference for scene-relative object motion are predictive of the animals' perceptual decisions and that training a linear decoder to detect object motion based on MT responses largely reproduces our main empirical results. We also show that selectively decoding neurons with incongruent tuning yields better performance than decoding congruent neurons. Our findings establish a novel mechanism for detecting moving objects during self-motion, thus revealing a sensory substrate for a specific form of causal inference. Because this mechanism relies on sensitivity to local discrepancies between disparity and motion parallax cues, it allows detection of object motion without the need for more complex computations that discount the global flow field. Thus, this local mechanism may be relatively economical for the nervous system to implement and likely provides a complementary approach to mechanisms for computing scene-relative object motion based on optic flow parsing (*Rushton and Warren, 2005*; *Warren and Rushton, 2008*; *Warren and Rushton, 2009b*, a; *Layton and Fajen, 2016b*; *Niehorster and Li, 2017*; *Layton and Niehorster, 2019*; *Layton and Fajen, 2020*; *Peltier et al., 2020*).

## Results

We recorded from 123 well-isolated single neurons in area MT of two macaques that were trained to perform an object motion detection task during self-motion (53 neurons from monkey 1 [M1] and 70 from monkey 2 [M2]). We begin by describing the task and behavioral data, followed by analysis of the responses of isolated MT neurons during this task. Finally, we demonstrate that a simple linear decoder trained to perform the task based on responses of our MT population can recapitulate our main findings.

### Stimulus configuration and behavioral task

During neural recordings, monkeys viewed a display consisting of two square planar objects that were defined by random dot patterns (*Figure 1A, B*; *Figure 1—figure supplement 1*; *Video 1*; see Methods for details). The animal viewed these objects while being translated (0.5 Hz modified sinusoid, see Methods) along an axis in the fronto-parallel plane which corresponded with the preferred-null motion axis of the neuron under study. In the base condition of the task with no cue conflict between depth from disparity and motion parallax, both objects were simulated to be stationary in the world, such that their image motion was determined by the self-motion trajectory and the location of the objects in depth. When the objects were stationary in the world, their depth defined by motion parallax and disparity cues was the same, hence the difference in depth between the two cues was zero ($\Delta$Depth = $d_{MP}$ − $d_{BD}$ = 0).

In other conditions ($\Delta$Depth ≠ 0), one of the objects was stationary in the world while the second 'dynamic' object moved in space such that its depth defined by motion parallax, $d_{MP}$, was not consistent with its depth defined by binocular disparity, $d_{BD}$ (*Figure 1B*, *Figure 1—figure supplement*

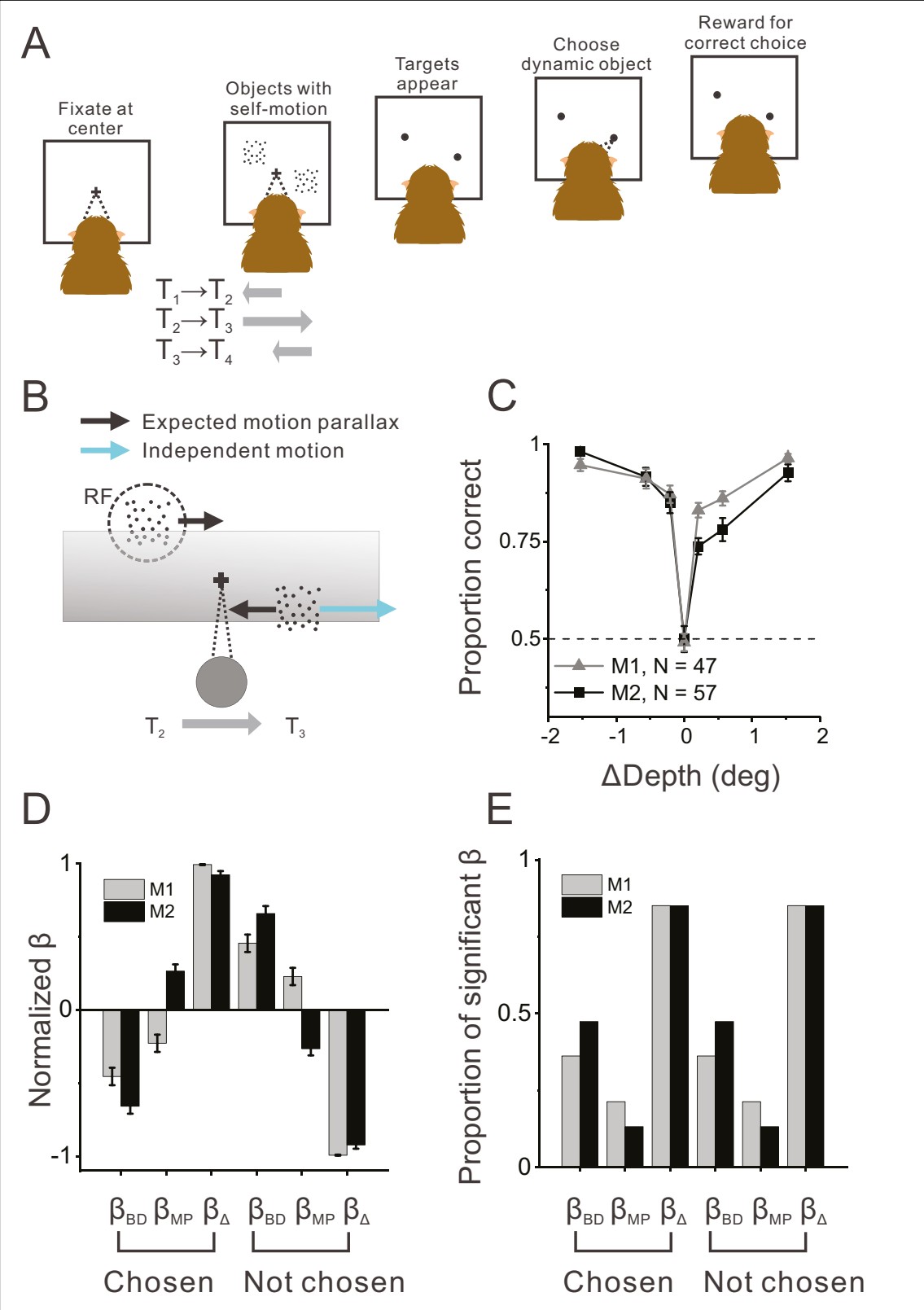

**Figure 1.** Object detection task and behavior. (**A**) Schematic illustration of the moving object detection task. Once the animal fixated on a center target, objects were presented while the animal experienced self-motion. Saccade targets then appeared at the center of each object, and the animal indicated the dynamic object (moving relative to the scene) by making a saccade. (**B**) Schematic illustration of stimulus generation from behind and above the observer. A stationary far object that lies within the neuron's receptive field (RF, dashed circle) has rightward image motion when the observer moves

*Figure 1 continued on next page*

*Figure 1 continued*

to the right. The other (dynamic) object moves rightward independently in space (cyan arrow) such that the object's net motion suggests a far depth while binocular disparity cues suggest a near depth. Gray shaded region indicates the display screen; cross indicates the fixation point. (**C**) Average behavioral performance across recording sessions for each animal (n=47 sessions from monkey 1 [M1] and n=57 sessions from monkey 2 [M2], excluding two sessions for which the standard set of ΔDepth values was not used). Error bars denote 95% CIs. (**D**) Normalized regression coefficients for depth from disparity (β$_{BD}$), depth from motion parallax (β$_{MP}$), and ΔDepth (β$_Δ$) are shown separately for chosen locations and not-chosen locations (see text for details). Gray and black bars denote data for M1 (n=44) and M2 (n=53), respectively. (**E**) Proportion of fits for which each regression coefficient was significantly different from zero (alpha = 0.05). Format as in panel D.

The online version of this article includes the following figure supplement(s) for figure 1:

**Figure supplement 1.** Visual display and motion trajectories.

**Figure supplement 2.** Results from a control experiment including monocularly presented objects.

**Figure supplement 3.** Behavioral performance in the more generalized task with four objects.

---

*1B*; see Methods for details). As a result of this cue conflict between disparity and motion parallax, the dynamic object should appear to be moving in the world based on previous work in humans (*Rushton et al., 2007*). As ΔDepth becomes greater in magnitude, it should be easier for the animal to correctly determine which object is the dynamic object. Animals indicated their decision by making a saccade to one of two targets that appeared at the locations of the two objects at the end of the trial (*Figure 1A*). Critically, due to the experimental design (see Methods for details), animals could not simply detect the dynamic object based on its retinal image velocity since the stationary object(s) in the display also moved on the retina due to self-motion combined with depth variation.

Average psychometric functions for the two animals across 104 recording sessions are shown in *Figure 1C*. As expected, the animals perform at chance when ΔDepth = 0, and their percent correct increases with the magnitude of ΔDepth. This demonstrates that monkeys can perform the task as expected from human behavioral work (*Rushton et al., 2007*). Furthermore, we found that performance was near chance levels in a control experiment without binocular disparity cues (*Figure 1—figure supplement 2*), as also expected from previous work (*Rushton et al., 2007*).

The ranges of depths of the stationary and dynamic objects were overlapping but not identical (see Methods). To determine whether the animals primarily made their decisions based on ΔDepth and not based upon the individual depths specified by disparity or motion parallax, we performed a logistic regression analysis to determine how animals perceptually weighted depth from motion parallax ($|d_{MP}|$), depth from binocular disparity ($|d_{BD}|$), and the magnitude of ΔDepth ($|d_{MP} - d_{BD}|$; see Methods for details). Results show that animals primarily weighted the |ΔDepth| cue to make their decisions (*Figure 1D, E*), although there were small contributions from the individual depth cues. We initially trained each animal to perform the task with four objects present in the display (three stationary objects and one dynamic object), as well as three different pedestal depths, to make it more difficult for animals to rely on d$_{MP}$ or d$_{BD}$. Indeed, we found that the logistic regression weights were also strongly biased in favor of |ΔDepth| in the four-object version of the task (*Figure 1—figure supplement 3D, E*). To increase the number of stimulus repetitions we could perform during recording experiments, we simplified the task to the two-object case.

## Congruency of depth preferences and responses to dynamic objects

We measured the tuning of well-isolated MT neurons for depth defined by either binocular disparity or motion parallax cues, as described previously (*Nadler et al., 2013*, see also Methods). Receptive fields (RFs) and direction preferences of the population of MT neurons are summarized in *Figure 2—figure supplement 1*. *Figure 2A* shows data for a typical 'congruent' cell, which prefers near depth defined by both disparity and motion parallax cues (see Methods for definition of congruent and opposite cells). Note that motion parallax stimuli are presented monocularly, such that selectivity for depth from motion parallax cannot be a consequence of binocular cues. In contrast, *Figure 2B, C* show data for two examples of 'opposite' cells that prefer near depths defined by motion parallax and moderate far depths defined by binocular disparity. Such neurons would, in principle, respond more strongly to some stimuli with discrepant disparity and motion parallax cues. Note that, for all of the example cells in *Figure 2A–C*, responses to binocular disparity are substantially greater than responses to motion parallax. This is mainly because binocular disparity tuning was measured with

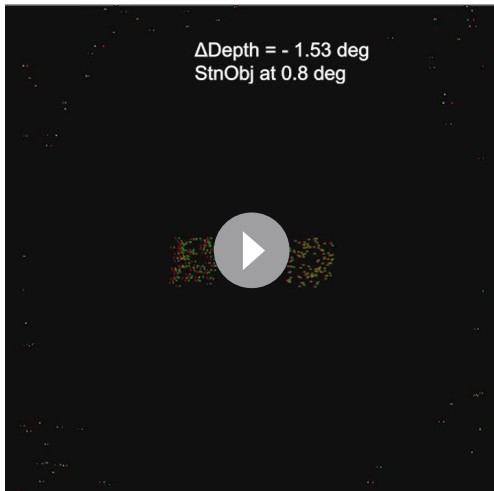

**ΔDepth = - 1.53 deg**
**StnObj at 0.8 deg**

**Video 1.** Visual stimuli used in the dynamic object detection task. Examples of visual stimuli in the two-object task, assuming that the receptive field of a neuron is located on the horizontal meridian. The video shows a sequence of seven stimuli, which are sorted by their ΔDepth values (ΔDepth = −1.53, −0.57, −0.21, 0, 0.21, 0.57, and 1.53 deg). The depth of the stationary object in each stimulus is labeled and was chosen randomly. In the actual experiment, the fixation target was stationary in the world, and the motion platform moved the animal and screen sinusoidally along an axis in the fronto-parallel plane (here a horizontal axis). Thus, the video shows the scene from the viewpoint of the moving observer. The stimulus sequences are equivalent to a situation in which the observer remains stationary, and the entire scene is translated in front of the observer. For each ΔDepth value, two full cycles of the stimulus are shown for display purposes; in the actual experiment, each trial consisted of just one cycle. During the second cycle of each stimulus in the video, the text label indicates whether the dynamic object was on the left or right side of the display. Red and green dots in the video denote the stereo half-images for the left and right eyes. Note that, without viewing the images stereoscopically and tracking the fixation target, it is generally not possible to determine the location of the dynamic object from the image motion on the display.

https://elifesciences.org/articles/74971/figures#video1

constant-velocity stimuli at the preferred speed, whereas the range of speeds used to measure depth tuning based on motion parallax is generally lower (and covaries with depth magnitude).

As done previously (*Nadler et al., 2008*; *Nadler et al., 2009*; *Nadler et al., 2013*; *Kim et al., 2015a*, *Kim et al., 2015b*; *Kim et al., 2017*), we quantified the depth-sign preference of each MT neuron using a depth-sign discrimination index (DSDI, see Methods), which takes on negative values for neurons with near preferences and positive values for neurons with far preferences. Across the population of 123 neurons, depth-sign preferences for motion parallax tended to be strongly biased toward near-preferring neurons, as reported previously (*Nadler et al., 2008*; *Nadler et al., 2013*), whereas depth-sign preferences for binocular disparity were rather well balanced (*Figure 3A*). Importantly, there are roughly equal numbers of neurons in the lower-left and upper-left quadrants of *Figure 3A*, indicating that congruent and opposite cells were roughly equally prevalent in our sample of MT neurons (see also *Nadler et al., 2013*). Thus, there are many opposite cells in MT that might respond selectively to dynamic objects over static objects.

*Figure 2D* shows responses of the example congruent cell (from *Figure 2A*) that were obtained during the object detection task. Responses to the stationary object (red) are plotted as a function of the depth values specified by motion parallax (which are necessarily equal to binocular disparity values for a stationary object). Responses to the dynamic object (blue) are plotted as a function of both depth defined by motion parallax (lower abscissa) and depth defined by disparity (upper blue abscissa). This allows the reader to determine the depth value for each cue that is associated with a dynamic object having a particular ΔDepth value. For this example congruent cell (*Figure 2D*), responses to stationary objects with large near depths substantially exceeded responses to any dynamic object.

A strikingly different pattern of results is seen for the example opposite cell in *Figure 2E*. In this case, there are a few dynamic objects for which the neuron's response (blue) clearly exceeds the response to stationary objects of all different depth values (red). More specifically, this incongruent cell responds most strongly to dynamic objects that have large near depths defined by motion parallax and depths near the plane of fixation (0 deg) as defined by binocular disparity. This pattern of results is expected from the individual tuning curves in *Figure 2B* and demonstrates that this opposite cell is preferentially activated by a subset of dynamic objects. The second example opposite cell in *Figure 2C, F* shows a generally similar pattern of results. For this cell, peak responses to stationary and dynamic objects are similar, but the neuron responds more strongly to dynamic objects over most of the stimulus range. Since we applied our ΔDepth manipulation around a fixed pedestal depth of

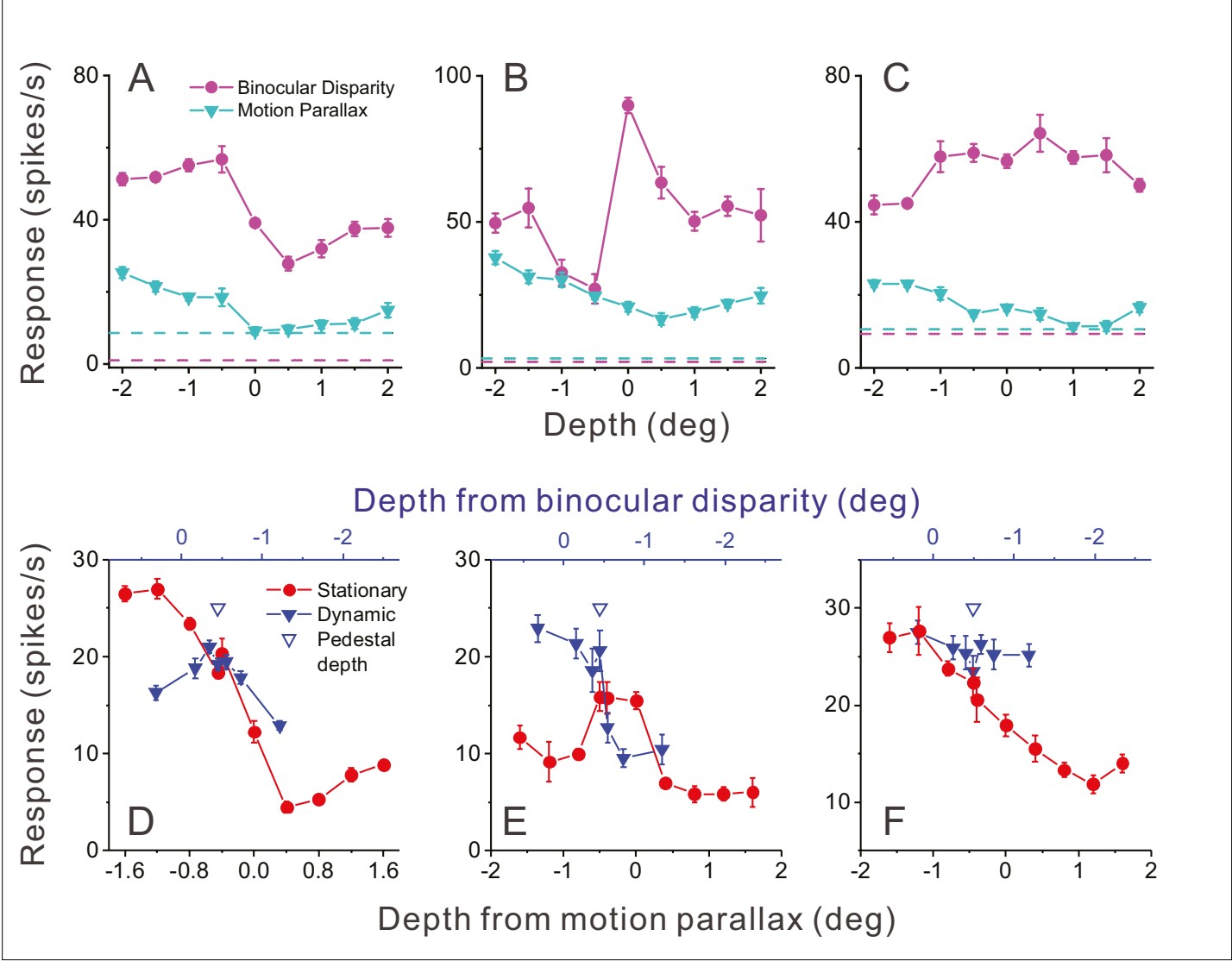

**Figure 2.** Responses of representative MT neurons. (**A**) Depth tuning curves for an example 'congruent' neuron preferring near depths based on both binocular disparity (magenta) and motion parallax (cyan) cues ($DSDI_{BD} = -0.81$; $DSDI_{MP} = -0.70$ [DSDI, depth-sign discrimination index]; $p<0.05$ for both, permutation test; correlation $R_{MP\_BD} = 0.76$, $p=0.016$). Dashed horizontal lines indicate baseline activity for each tuning curve. (**B**) Tuning curves for an example 'opposite' neuron preferring small far depths based on binocular disparity but preferring near depths based on motion parallax ($DSDI_{BD} = 0.41$; $DSDI_{MP} = -0.67$; $p<0.05$ for both, permutation test; $R_{MP\_BD} = -0.36$, $p=0.32$). (**C**) Another example opposite cell preferring far depths based on binocular disparity but near depths based on motion parallax ($DSDI_{BD} = 0.46$; $DSDI_{MP} = -0.56$; $p<0.05$ for both, permutation test; $R_{MP\_BD} = -0.73$, $p=0.025$). (**D**) Responses of the neuron in panel A to stationary objects (red) and dynamic objects (blue) during performance of the detection task. Stationary objects were presented at various depths (bottom abscissa). Dynamic objects generally have conflicts ($\Delta$Depth $\neq 0$) between depth from motion parallax (bottom abscissa) and binocular disparity (top abscissa). The pedestal depth at which $\Delta$Depth $= 0$ is shown as an unfilled blue triangle. (**E**) Responses during the detection task for the opposite cell of panel B. Format as in panel D. (**F**) Responses during detection for the neuron of panel C. Error bars in all panels represent s.e.m.

The online version of this article includes the following figure supplement(s) for figure 2:

**Figure supplement 1.** Distribution of receptive field (RF) properties.

–0.45 deg (to facilitate decoding, see Methods), we do not expect dynamic objects to preferentially activate every opposite cell. However, cells that are preferentially activated by dynamic objects should tend to be neurons with mismatched depth tuning for motion parallax and binocular disparity cues.

*Figure 3B* shows that this expected relationship holds across our population of MT neurons. The ratio of peak responses for dynamic:stationary objects is plotted as a function of the correlation coefficient, $R_{MP\_BD}$, between depth tuning curves for disparity and motion parallax. Neurons with $R_{MP\_BD} < 0$

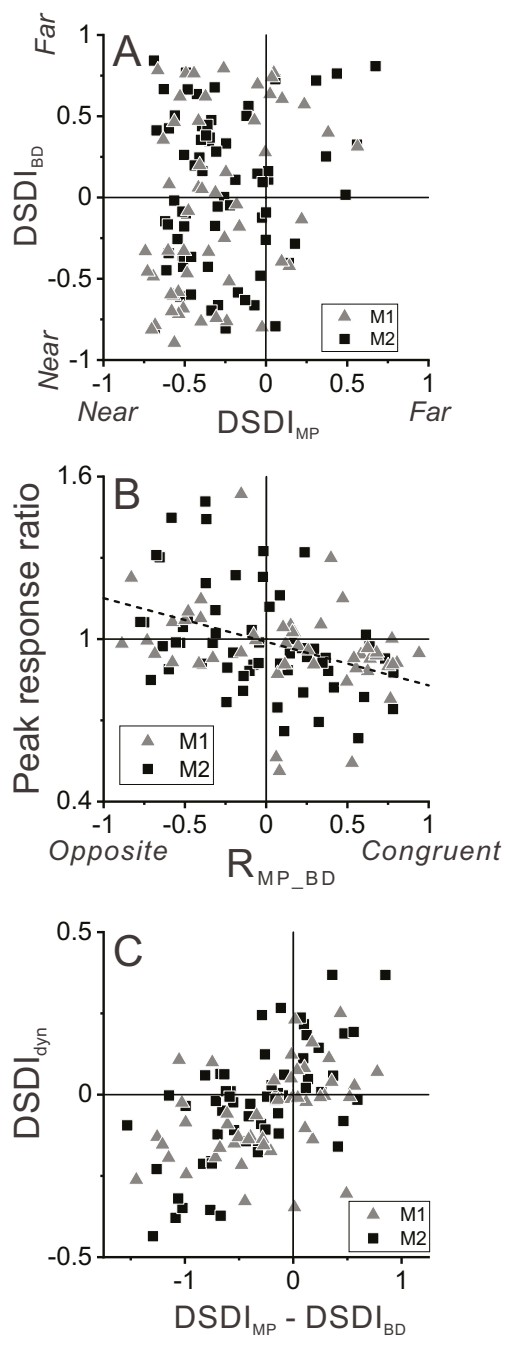

*Figure 3 continued*

The ordinate shows the ratio of peak responses for dynamic:stationary stimuli. The abscissa shows the correlation coefficient ($R_{MP\_BD}$) between depth tuning for motion parallax and disparity. Dashed line is a linear fit using type 2 regression (n=106; n=47 from M1 and n=59 from M2; sample includes all neurons for which we completed the detection task). (**C**) Population summary (n=106) of the relationship between the preference for ΔDepth of the dynamic object (as quantified by $DSDI_{dyn}$, see Methods) and the difference between $DSDI_{MP}$ and $DSDI_{BD}$.

(opposite cells) tend to have peak response ratios that lie in the upper-left quadrant, indicating that opposite cells tend to be preferentially activated by dynamic objects. In contrast, neurons with $R_{MP\_BD}$ > 0 (congruent cells) tend to have peak response ratios in the lower-right quadrant, indicating that they tend to be preferentially activated by stationary objects. Across the population, peak response ratio is significantly anti-correlated with $R_{MP\_BD}$ (n=106, Spearman rank correlation, R = −0.39, p=2.8×10⁻⁵), indicating that the hypothesized relationship between tuning congruency and response to scene-relative object motion is observed.

We further tested whether differences in depth tuning curves for binocular disparity and motion parallax can predict whether neurons prefer positive or negative ΔDepth values. Using responses to the dynamic object, we quantified each neuron's preference for positive/negative ΔDepth values using a variant of the DSDI metric, $DSDI_{dyn}$ (see Methods), and found that it is robustly correlated with the difference in DSDI values (ΔDSDI) computed from depth tuning curves for disparity and motion parallax (***Figure 3C***, R=0.54, p=2.7×10⁻⁹, n=106, Spearman correlation). Thus, selectivity for ΔDepth during the detection task is reasonably predictable from the congruency of depth tuning measured during a fixation task.

## Correlation with perceptual decisions

If neurons with mismatched depth tuning for disparity and motion parallax cues are selectively involved in detecting scene-relative object motion, we hypothesized that neurons that respond preferentially to dynamic objects should have responses that are more strongly correlated with perceptual decisions. To measure the correlation of neural activity with perceptual decisions, we took advantage of the fact that our design included a subset of trials in which both objects

**Figure 3.** Relationship between selectivity for moving objects and congruency between depth cues. (**A**) Population summary of congruency of depth tuning for disparity and motion parallax. The depth-sign discrimination index (DSDI) value for binocular disparity tuning ($DSDI_{BD}$) is plotted as a function of the DSDI value for motion parallax tuning ($DSDI_{MP}$) for each neuron (n=123). Triangles and squares denote data for monkey 1 (M1) (n=53) and monkey 2 (M2) (n=70), respectively. (**B**) Population summary of relationship between relative responses to dynamic and stationary objects as a function of depth tuning congruency.

were stationary in the world and were presented at the pedestal depth of –0.45 deg (*Figure 4A*). These conditions allowed us to quantify choice-related activity, for a fixed stimulus, by sorting responses into two groups: trials in which the monkey chose the object in the neuron's RF, and trials in which the monkey chose the object in the opposite hemi-field.

Data for an example neuron (*Figure 4B*) show somewhat greater responses when the monkey chose the object located in the neuron's RF. We quantified this effect by applying ROC analysis to the two choice distributions (see Methods for details), which yielded a detection probability (DP) metric. DP will be greater than 0.5 when responses are greater on trials in which the monkey reported that the stimulus in the RF was the dynamic object. Because DP is only computed from the subset of trials with ΔDepth=0, this measure need not have any relationship with a neuron's preference for dynamic vs. stationary stimuli. For the example neuron of *Figure 4B*, the DP value was 0.75, which is significantly greater than chance by permutation test (p=0.006, see Methods). Across a population of 92 neurons for which there were sufficient numbers of choices toward each stimulus (see Methods), the mean DP value of 0.56 was significantly greater than chance (p=6×10$^{-5}$, t(91) = 4.21, n=92, *t*-test) with 13 of 92 neurons showing individually significant DP values (*Figure 4C*, filled bars). All neurons with significant DP values had effects in the expected direction, with DP>0.5. In addition, the mean DP value was significantly greater than chance for each monkey individually (M1: n=39, mean=0.59, p=5.7×10$^{-4}$, t(38) = 3.76; M2: n=53, mean = 0.53, p=0.03, t(52) = 2.21, *t*-test).

*Figure 4* shows that many MT neurons have responses that are correlated with detection choices in the task. We hypothesized that neurons with DP>0.5 are more likely to be those that respond preferentially to dynamic objects over stationary objects. To obtain a signal-to-noise measure of each neuron's selectivity for dynamic vs. stationary objects, we again applied ROC analysis as illustrated for an opposite cell in *Figure 5A–C*. This neuron responded more strongly to dynamic objects than stationary objects across most of the depth range (*Figure 5A*). To quantify this selectivity, for each value of ΔDepth, responses were sorted into two groups: trials in which the dynamic object was in the RF, and trials in which the dynamic object was located in the opposite hemi-field and a stationary object was in the RF (regardless of the depth of the stationary object).

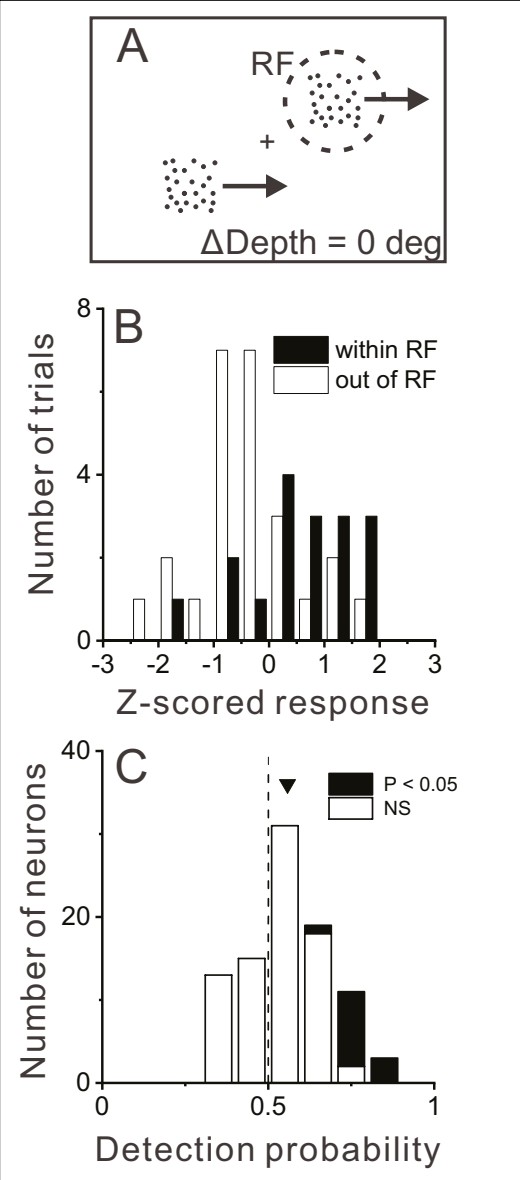

**Figure 4.** Relationship between MT responses and detection of object motion. (**A**) When ΔDepth = 0, two stationary objects at the pedestal depth had identical retinal motion and depth cues. Animals still were required to report one of the objects as dynamic. (**B**) To compute detection probability (DP), responses to the ΔDepth = 0 condition were z-scored and sorted into two groups according to the animal's choice. Filled and open bars show distributions of z-scored responses of an example MT neuron when the animal reported that the moving object was in and out of the receptive field (RF), respectively. (**C**) Distribution of DP values for a sample of 92 MT neurons, including all neurons tested in the detection task for which the animal made at least five choices in each direction (see Methods). Arrowhead shows the mean DP value of 0.56, which was significantly greater than 0.5 (p=6.0×10$^{-5}$, n=92, *t*-test).

The online version of this article includes the following

*Figure 4 continued*

figure supplement(s) for figure 4:

**Figure supplement 1.** Time courses of choice-related activity.

Thus, the ROC value computed for each ΔDepth value gave an indication of how well the neuron discriminated between that particular dynamic object and stationary objects of any depth. By convention, ROC values >0.5 indicate greater responses for a dynamic object in the RF.

Results of this analysis for the example opposite cell (*Figure 5B*) show that ROC values were greater than 0.5 for all ΔDepth ≠ 0; thus, this neuron reliably responded more strongly to dynamic objects than to stationary objects. To obtain a single metric for each neuron, we simply averaged the ROC metrics for each non-zero ΔDepth value, yielding a neurometric performance (NP) value of 0.78 for this neuron. The corresponding DP value for this neuron was 0.77 (*Figure 5C*, p=0.0015, permutation test), indicating that this neuron shows both strong selectivity for dynamic objects when ΔDepth ≠ 0 and stronger responses when the animal reports a dynamic object in the RF when ΔDepth = 0.

Data for an example congruent cell (*Figure 5D–F*) show a very different pattern of results. This neuron generally responds more strongly to stationary objects of any depth than to dynamic objects (*Figure 5D*). As a result, ROC values are consistently <0.5 when comparing responses to dynamic vs. stationary objects in the RF (ΔDepth ≠ 0, *Figure 5E*), yielding an NP value of 0.23. The corresponding DP value for this neuron (*Figure 5F*) was 0.39 (p=0.26, permutation test), indicating that it responded slightly more to ambiguous stimuli when the monkey reports that the object in the RF was stationary. Thus, the data from these two examples neurons support the hypothesis that neurons with preferences for dynamic objects are selectively correlated with perceptual decisions.

To examine whether this hypothesis holds at the population level, we plotted the DP value for each neuron against the corresponding NP value. These two metrics, which are computed from completely different sets of trials (ΔDepth = 0 for DP; ΔDepth ≠ 0 for NP), are strongly correlated (*Figure 5G*, $R$=0.47, p=3.2×10⁻⁶, n=92, Spearman rank correlation) such that neurons with DP values substantially greater than 0.5 tend to be neurons that are selective for dynamic objects (NP >0.5). In addition, we observed a significant positive correlation for each animal individually (M1: n=39, $R$=0.59, p=7.9×10⁻⁵; M2: n=53, $R$=0.33, p=0.015, Spearman correlation). It is also worth noting that all neurons with large DP values (>0.7) also have NP values substantially greater than 0.5. Thus, the MT neurons that most strongly predict decisions to detect the dynamic object (on ambiguous trials) are those with a consistent preference for dynamic objects.

It is worth noting that the distribution of NP values in *Figure 5G* is biased toward values >0.5; indeed, the mean NP value (0.56) is significantly greater than 0.5 (one-sample *t*-test, $t(105)$ = 4.98, and p=2.5×10⁻⁶). This effect likely arises due to the distribution of stimulus values involved in the dynamic object condition. Because neurons were generally tested with a pedestal depth of –0.45 deg, the depth values for both disparity and motion parallax tend to be mostly negative for the dynamic object condition (see blue and black x-axes in *Figure 2D–F*). This bias toward negative (near) depth values of dynamic objects, combined with the fact that most neurons have a near preference for depth from motion parallax (*Figure 3A*), means that many neurons (including congruent cells) tend to have mean responses to dynamic objects that are greater than the mean response to stationary objects (e.g. *Figure 2D*). This asymmetry leads to a mean NP value >0.5.

How is the result of *Figure 5G* related to neurons' preference for dynamic relative to stationary objects? *Figure 5—figure supplement 1A* shows that NP is robustly correlated with peak response ratio ($R$=0.43, p=4.5×10⁻⁶, Spearman rank correlation). Neurons with peak response ratios substantially greater than unity almost always have NP >0.5. However, some neurons with peak response ratios near unity also have high NP values (including the neuron in *Figure 2C, F*). In contrast, we did not find a significant correlation between DP and peak response ratio (*Figure 5—figure supplement 1B*: $R$=0.12, p=0.25). A main reason for this appears to be that there is a cluster of neurons with large peak response ratios that have DP values near 0.5. These neurons generally have tuning properties similar to the example cell in *Figure 2B, E*. While incongruent tuning creates a clear preference for dynamic objects for these neurons, that preference is limited to a narrow range of depth values, such that NP values tend to be only modestly >0.5. To achieve high NP values, neurons need to have a consistent preference for dynamic objects (e.g. *Figures 2F and 5A*). Thus, what seems to be crucial for producing high DP values is that a neuron consistently prefers dynamic objects over the range

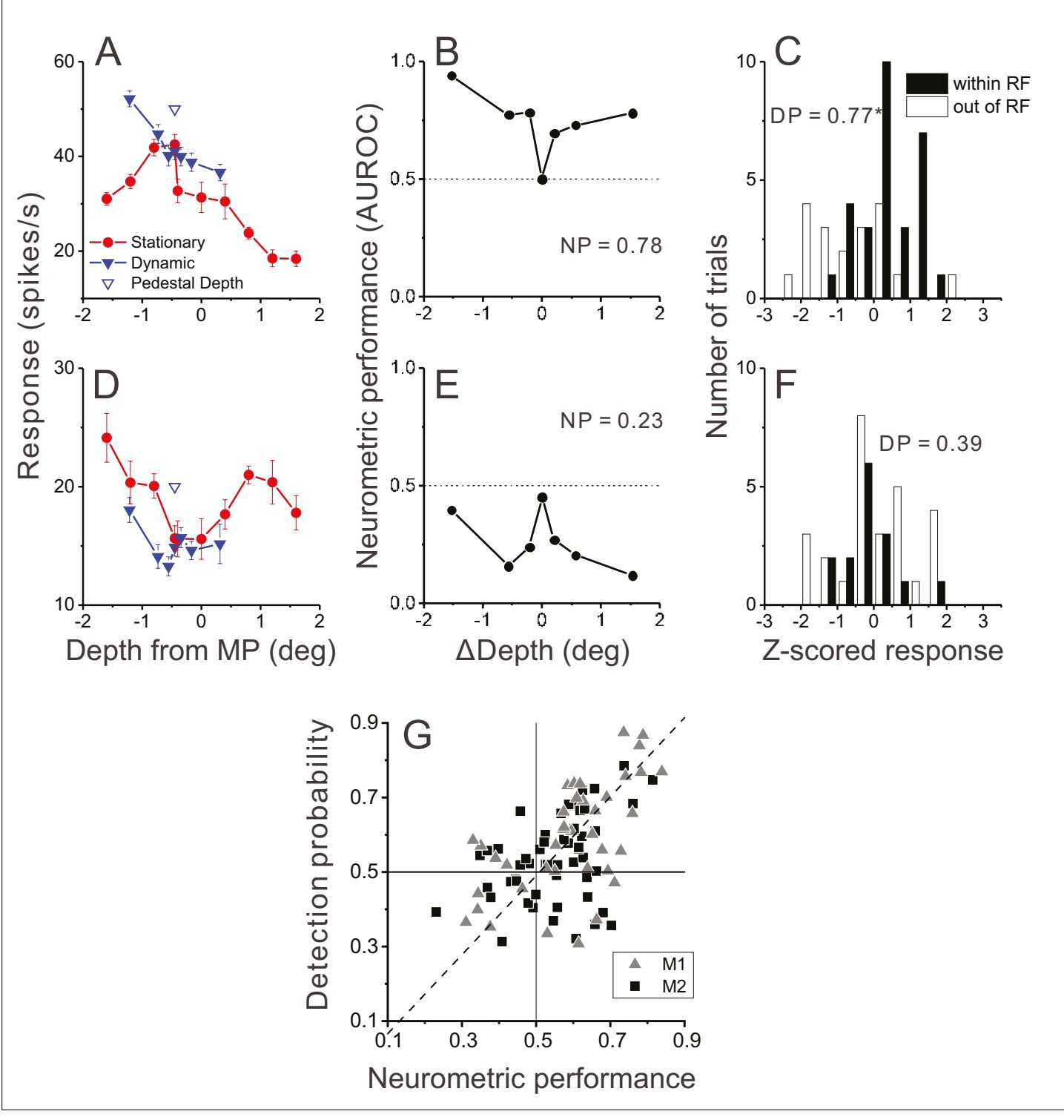

**Figure 5.** Relationship between detection probability (DP) and neurometric performance for dynamic objects. (**A**) Responses of an example opposite neuron to dynamic and stationary objects during the detection task. Format as in *Figure 2D*. (**B**) ROC values comparing responses to a dynamic object at each value of ΔDepth with responses to stationary objects, for the neuron of panel A. Neurometric performance (NP = 0.78 for this neuron) is defined as the average ROC area for all ΔDepth ≠ 0. (**C**) Distribution of z-scored responses sorted by choice for the same neuron as in panels A,B. Format as in *Figure 4B*. (**D–F**) Data from an example congruent cell, plotted in the same format as panels A–C. (**G**) Relationship between DP and NP for a population of 92 MT neurons. Dashed line: linear fit using type 2 regression (slope = 1.06, slope CI = [0.80 1.48]; intercept = –0.04, intercept CI = [–0.31 0.11]).

The online version of this article includes the following figure supplement(s) for figure 5:

**Figure supplement 1.** Relationships between preference for dynamic objects, neurometric performance (NP), and detection probability (DP).

tested. This explains the robust correlation between DP and NP (*Figure 5G*) and the weak correlation between DP and peak response ratio (*Figure 5—figure supplement 1B*). Thus, incongruent tuning tends to lead to a preference for dynamic objects (high peak response ratio, *Figure 3B*), but does not always lead to high DP values (*Figure 5—figure supplement 1B*).

We also examined the time course of choice-related activity and found that it appeared within a few hundred ms after the onset of self-motion (*Figure 4—figure supplement 1*). This choice-related activity was largely sustained throughout the rest of the stimulus period, even when motion of the object was in the anti-preferred direction.

## Decoding model

The results described above suggest that perceptual detection of dynamic objects might be driven by the activity of MT neurons with depth tuning curves that make them respond preferentially to dynamic objects. To further probe this hypothesis, we trained a simple linear decoder to detect dynamic objects based on simulated responses of a population of neurons that is closely based on our data, and we examined whether performance of the decoder shows a similar relationship between DP and NP (see Methods for details).

In the simulation (as in the experiments), the dynamic object could appear in either the right or left hemi-field, and the decoder was trained to report the location of the dynamic object. For each neuron

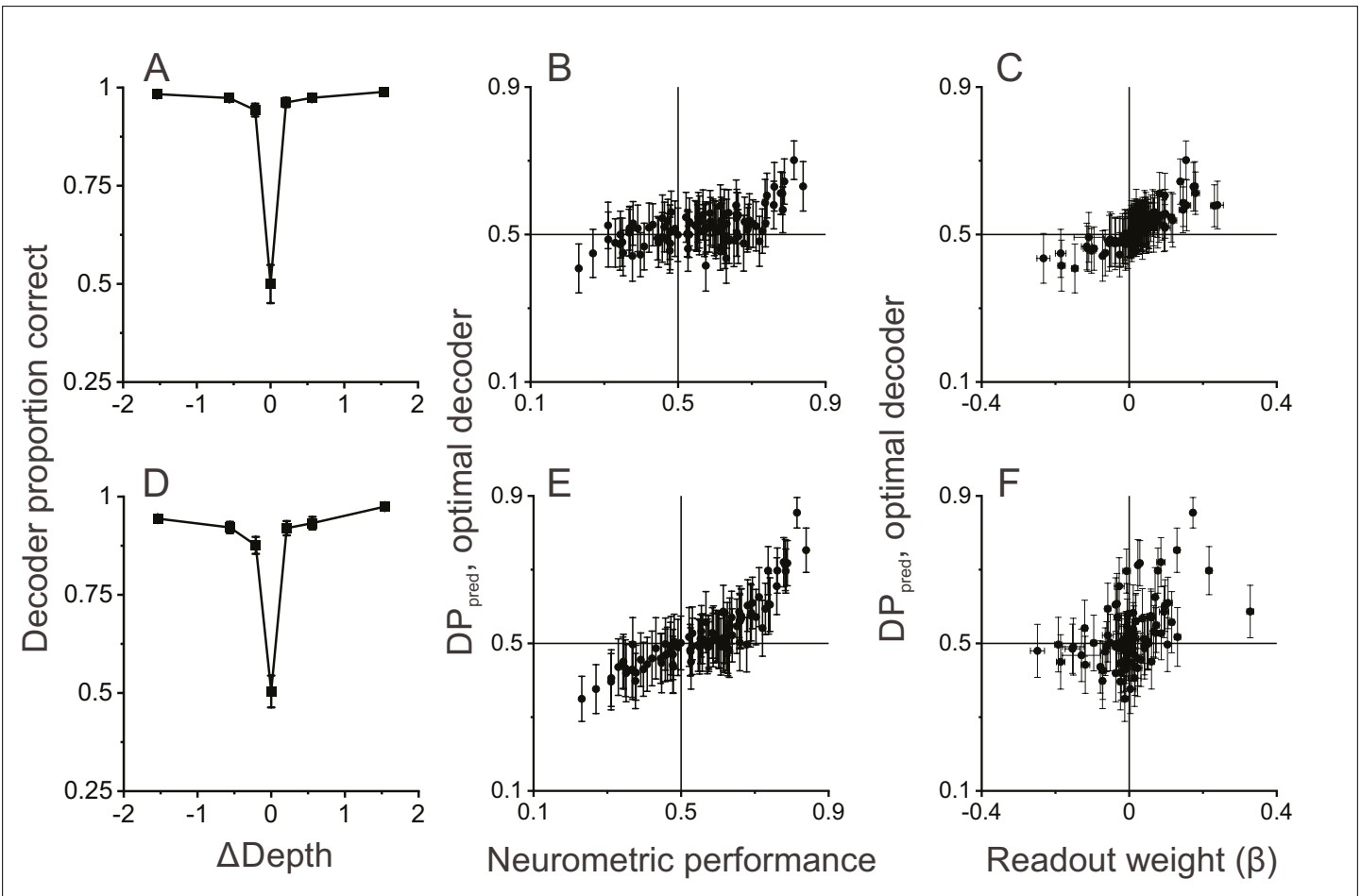

**Figure 6.** Linear decoding reproduces the relationship between detection probability (DP) and neurometric performance (NP). (**A**) Performance of a linear decoder that was trained to detect moving objects based on simulated population responses with independent noise (see Methods for details). Error bars represent 95% CIs (n=100 simulations). (**B**) Neural responses were sorted by the output of the decoder to compute a predicted DP (DP$_{pred}$) for each unit in the simulated population (n=97, including all neurons recorded in the detection task using identical ΔDepth and stationary depth values, see Methods). DP$_{pred}$ is plotted as a function of the measured NP for each neuron. Error bars represent 95% CIs (n=100 simulations). (**C**) Relationship between DP$_{pred}$ and the readout weight (β) for each unit in the decoded population. Error bars represent 95% CIs. (**D–F**) Analogous results for a decoder that was trained based on population responses with modest correlated noise (see text and Methods for details). Format as in panels A-C.

in the population, responses were simulated to have the same mean and SD as empirically measured responses. Since neurons were recorded separately and we could not measure correlated noise, we simulated responses based on either independent noise or correlated noise (see Methods for details).

The decoder was trained to report the location of the dynamic object based on simulated population responses from the subset of trials for which ΔDepth ≠ 0. The trained decoder was then used to predict responses for the completely ambiguous (ΔDepth = 0) trials in which identical objects were presented in both hemi-fields. Responses to ambiguous trials were then sorted according to the decoder output to compute predicted DP values (DP_pred) for each neuron in the population.

We first compared DP_pred with NP values for simulations in which all neurons were assumed to have independent noise. This decoder performs very well based on a sample of 97 MT neurons (*Figure 6A*, see Methods for selection criteria), indicating that there is extensive information available in a moderately sized sample of MT neurons. We find a significant positive correlation between DP_pred and NP (*Figure 6B*, $R=0.53$, $p=4.9\times10^{-8}$, $n=97$, Spearman correlation) in this simulation, consistent with the empirical observations of *Figure 5G*. The decoding weights provide an indication of how neurons with different properties contribute to the classification outcomes. With independent noise, we find a strong relationship (*Figure 6C*, $R=0.92$, $p<1\times10^{-15}$, $n=97$, Spearman correlation) between DP_pred and decoding weights, with positive readout weights being associated with DP_pred values greater than 0.5.

While the relationship between DP_pred and NP in *Figure 6B* has a positive slope, DP_pred values tend to be substantially closer to 0.5 than the values observed experimentally (*Figure 5G*). However, this is not surprising given that neurons in this simulation were assumed to have independent noise. It is well established that neurons in MT exhibit correlated noise (e.g. *Zohary et al., 1994*; *Huang and Lisberger, 2009*) and that choice-related activity is expected to be stronger in the presence of correlated noise (*Britten et al., 1996*; *Shadlen et al., 1996*; *Haefner et al., 2013*; *Gu et al., 2014*; *Pitkow et al., 2015*). Thus, we also simulated responses with a moderate level of correlated noise (median $R_{noise} = 0.15$, see Methods for details), which had little impact on decoder performance (*Figure 6D*). In the presence of correlated noise, DP_pred values show a greater spread around 0.5 and are much more strongly correlated with NP values (*Figure 6E*, $R=0.89$, $p<3\times10^{-16}$, $n=92$, Spearman correlation). While correlated noise enhances the relationship between DP_pred and NP, it also weakens the relationship between DP_pred and decoding weights (*Figure 6F*, $R=0.46$, $p=3.4\times10^{-6}$, $n=92$, Spearman correlation), as expected from theoretical studies (*Haefner et al., 2013*).

These simulations show that our main experimental finding is recapitulated by a simple linear decoder that is trained to distinguish between dynamic and static objects based on MT responses. Note, however, that we have not attempted to find parameters of our decoding simulations that would best match the empirical data (*Figure 5G*). This would almost certainly be possible, but we do not feel that it is a worthwhile exercise given that we would have to make assumptions about the structure of correlated noise that we cannot sufficiently constrain.

To further assess whether neurons with incongruent tuning for disparity and motion parallax can provide a greater contribution to detecting scene-relative object motion, we performed additional decoding simulations after dividing the population into subgroups based on peak response ratio. Four subgroups were defined: lowest third, middle third, highest third, and a random subsample of the same size from all neurons. *Figure 7A* shows that decoding performance is significantly greater for the subgroup of neurons with the highest peak response ratios, as compared with the low and middle subgroups (error bars denote 95% CIs). We also performed a similar analysis after dividing the population based on the correlation between depth tuning from disparity and motion parallax cues, $R_{MP\_BD}$. This revealed parallel results in which neurons with the most negative values of $R_{MP\_BD}$ achieved significantly greater decoding performance (proportion correct: 0.88±0.008, mean ±95% CIs) than neurons with intermediate (0.82±0.01) or high (0.84±0.009) values of $R_{MP\_BD}$.

We further examined whether detection probabilities predicted by the decoder depend on tuning congruency, using the same subgroups based on peak response ratio. We find that DP_pred values from these decoding simulations are greatest for neurons with the highest peak response ratios, and not significantly above chance for neurons with the lowest peak response ratios (*Figure 7B*, see caption for details). These results demonstrate that neurons with incongruent tuning carry enhanced information for detecting moving objects during self-motion.

Together, these decoding simulations demonstrate that a population of MT neurons with the depth tuning properties that we have described could be utilized to detect scene-relative object

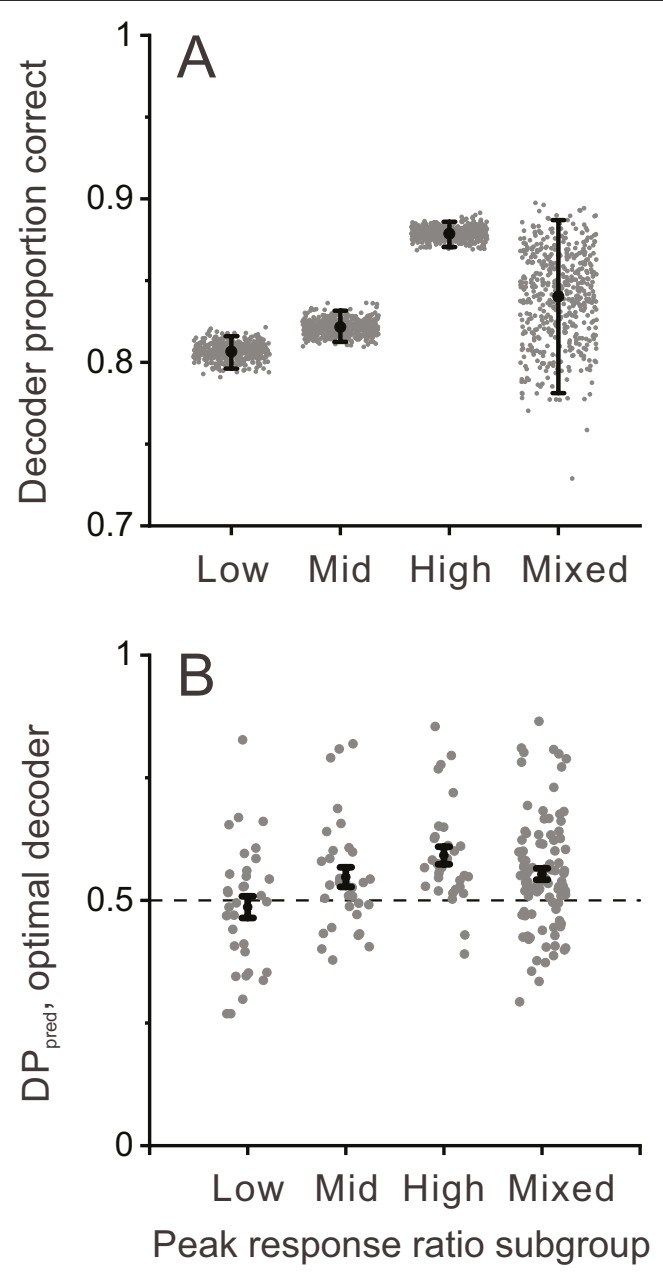

**Figure 7.** Decoder performance depends on selectivity for dynamic objects. Decoding was performed separately for three subgroups of MT neurons based on their peak response ratios: lowest third (n=33), middle third (n=32), and highest third (n=32) of peak response ratio values. In addition, decoding was performed for mixed subgroups of the same size (n=32) that were selected randomly from the population. (**A**) Decoder performance for each subgroup based on peak response ratio. For each subgroup, 500 decoding simulations were performed (gray dots) using different samples of correlated noise, as described in Methods. For the mixed subgroup, each of the 500 simulations also involved drawing (without replacement) a new random subset of 32 neurons from the population. Black filled symbols denote mean proportion correct across the 500 simulations, and error bars represent 95% CIs. (**B**) Predicted detection probability ($DP_{pred}$) for each neuron in each subgroup (gray symbols), along with mean $DP_{pred}$ values (black symbols, error bars denote s.e.m). For the mixed subgroup, data are shown for all 97 neurons since each neuron was included in many different subsamplings. In this case, gray data points represent the mean $DP_{pred}$ value for all random subsamplings (165, on average) that included each neuron. Median $DP_{pred}$ was not significantly different from 0.5 for the lowest third subgroup (p=0.59, n=33), was marginally significant for the middle third subgroup (p=0.043, n=32), and was significantly greater than 0.5 for the highest third (p=2.2×10$^{-5}$, n=32) and mixed (p=3.2×10$^{-5}$, n=97) subgroups (signed-rank tests comparing median $DP_{pred}$ with 0.5).

motion, and that such a read-out could produce the relationship between DP and NP that we have observed empirically. The simulations further indicate that neurons with larger peak response ratios provide better decoding performance, consistent with the idea that incongruent tuning is adaptive for detecting object motion during self-motion.

## Discussion

We find that neurons having incongruent depth tuning for binocular disparity and motion parallax cues often respond more strongly to objects that move in the world than to stationary objects. Moreover, neurons with a consistent preference for dynamic objects tend to more strongly predict perceptual decisions regarding object motion relative to the scene. While it has been established that humans can detect object motion based on cue conflicts between binocular disparity and motion parallax (*Rushton et al., 2007*), in the absence of other cues to object motion, the neural basis of this capacity has remained unknown. Our findings establish a simple neural mechanism for detecting moving objects, which can be computed locally and is complementary to flow-parsing mechanisms that involve more global computations (discussed further below). In addition, our findings establish another important function for neurons with mismatched tuning for multiple stimulus cues, building on recent studies (*Kim et al., 2016b*; *Goncalves and Welchman, 2017*; *Sasaki et al., 2017*; *Sasaki et al., 2019*; *Zhang et al., 2019a*). Our task involves a form of causal inference (*Körding et al., 2007*; *Shams and Beierholm, 2010*), and our findings support the idea that a sensory representation consisting of a mixture of congruent and opposite cells provides a useful sensory substrate for causal inference (*Rideaux et al., 2021*). To our knowledge, these findings provide the first empirical evidence for a specific contribution of opposite neurons to perceptual inference about causes of sensory signals.

### Comparison to other types of mechanisms for detecting object motion

In many instances, scene-relative object motion produces components of image motion that differ clearly in velocity or timing from the background optic flow at the corresponding location. Human observers can detect object motion when there are sufficient differences in local direction of motion between object and background (*Royden and Connors, 2010*). Humans can also detect object motion based on local differences in speed when there are sufficient depth cues (*Rushton et al., 2007*; *Royden et al., 2016*) or when the image speed of an object is outside the range of background speeds in a particular task context (*Royden and Moore, 2012*).

*Royden and Holloway, 2014* have shown that a model built on MT-like operators with surround suppression can effectively detect object motion when there are sufficient directional differences between object and background motion, or when object speed is outside the range of background speeds. However, such a model would not be able to detect object motion under task conditions like ours or those of *Rushton et al., 2007*, because our dynamic object had the same motion axis as the stationary distractors and because the speeds of our dynamic objects were well within the range of speeds of stationary objects. More recently, *Royden et al., 2015* have added disparity-tuned operators to their model, which allow detection of object motion even when it is aligned with background flow lines. This model computes local differences in response of separate velocity and disparity-tuned operators. It then applies an arbitrary threshold to detect cases for which there are differences and identifies these as possible object motion. While this model shows that differences in signals related to motion and disparity can be used to identify object motion in more general cases, it does not provide a biologically plausible neural mechanism.

Our findings demonstrate a key, and apparently thus far unappreciated, neural mechanism for identifying local discrepancies between binocular disparity and motion parallax cues that accompany moving objects, even in difficult cases for which there are no local differences in the direction or timing of image motion. The activity of MT neurons with incongruent depth tuning for motion parallax and disparity provides a critical signal about these local discrepancies. Moreover, our simulations indicate that these signals can be easily read out by a linear decoder to detect object motion during self-motion.

## Relationship to flow-parsing mechanism for computing scene-relative object motion

A more general approach to computing scene-relative object motion during self-motion is flow parsing, in which global patterns of background motion related to self-motion are discounted (i.e. subtracted off) such that the remaining signal represents object motion relative to the scene (*Rushton and Warren, 2005*). Several studies *Warren and Rushton, 2007*; *Warren and Rushton, 2008*; *Warren and Rushton, 2009a*; *Warren et al., 2012*; *Foulkes et al., 2013*; *Rushton et al., 2018* have provided strong behavioral support for the flow-parsing hypothesis in humans, including some which suggest strongly that it involves a global motion process (*Warren and Rushton, 2009b*). In addition, a recent study has demonstrated flow parsing in macaque monkeys (*Peltier et al., 2020*). If flow parsing completely discounts background motion due to self-motion, then the computations for detecting scene-relative object motion would be greatly simplified and would be essentially the same as when there is no self-motion. However, flow parsing alone may not be sufficient to detect scene-relative object motion. Recent evidence (*Niehorster and Li, 2017*; *Peltier et al., 2020*) indicates that the gain of flow parsing can be well below unity, such that background motion is only partially discounted. In this case, the output of a flow-parsing mechanism may not be sufficient to detect scene-relative object motion, and a mechanism such as we found in area MT would be valuable.

An advantage of our proposed mechanism over flow parsing is that it does not require estimation of the global flow field, nor a complicated mechanism (*Layton and Fajen, 2016b*; *Layton and Fajen, 2020*) for implementing the flow-parsing computation at each location in the visual field. Thus, our proposed mechanism may provide a valuable complement to flow parsing. On the other hand, the mechanism that we have described clearly does not obviate the need for flow-parsing mechanisms. Unlike our local mechanism, flow parsing does not require binocular disparity signals to operate. Furthermore, flow parsing allows for the estimation of scene-relative object velocity, rather than just facilitating object motion detection. Thus, our results do not discount the contributions of center-surround or flow-parsing mechanisms to computation of object motion, nor the contributions of mechanisms that may rely on multisensory signals (*Kim et al., 2016b*). Rather, our findings provide evidence of an additional complementary mechanism that is likely to be synergistic. Indeed, since our stimuli involved peripheral background dots (*Figure 1—figure supplement 1A*), it is possible that our task also engaged flow-parsing mechanisms to some extent.

We have recently demonstrated flow parsing in macaque monkeys (*Peltier et al., 2020*), including the observation that optic flow in one visual hemi-field can influence perception of object direction in the opposite hemi-field, as demonstrated previously in humans (*Warren and Rushton, 2009b*). These observations imply a global contribution to flow parsing, as might be implemented through feedback from the dorsal subdivision of the medial superior temporal area (MSTd) (*Layton and Fajen, 2016b*; *Layton and Fajen, 2020*). Indeed, in ongoing work, we have observed that flow parsing modulates responses of MT neurons (unpublished). In contrast, the mechanism described here can be computed locally within each portion of the visual field. Thus, we can speculate that the current findings and flow parsing involve distinct neural mechanisms in area MT.

Our findings establish the first direct (albeit correlational) evidence for a neural mechanism that is involved in perceptual dissociation of object and self-motion. However, the broader problem is more complex, as it is also necessary to flexibly compensate for self-motion to compute object motion in different coordinate frames, such as head-centered or world-centered reference frames (*Fajen et al., 2013*; *Sasaki et al., 2020*). Some of these computations are likely to also rely on non-visual signals including vestibular signals about self-motion (*Fajen and Matthis, 2013*; *Fajen et al., 2013*; *Dokka et al., 2015a*; *Dokka et al., 2015b*; *Sasaki et al., 2017*; *Dokka et al., 2019*; *Sasaki et al., 2020*). Thus, the mechanism proposed here is one part of a larger set of neural computations that remain to be fully understood.

## Functional roles of area MT and computational roles of opposite cells

Area MT has traditionally been considered to hold a retinotopic representation of retinal image motion. Many studies still make this assumption, despite the fact that MT is known to be modulated by attention (*Treue and Maunsell, 1996*; *Treue and Maunsell, 1999*; *Martínez-Trujillo and Treue, 2002*; *Womelsdorf et al., 2008*; *Lee and Maunsell, 2010*), eye movements (*Newsome et al., 1988*; *Bremmer et al., 1997*; *Inaba et al., 2007*; *Chukoskie and Movshon, 2009*; *Nadler et al., 2009*;

*Inaba et al., 2011*; *Kim et al., 2017*), and stimulus expectation (*Schlack and Albright, 2007*). Our previous work has shown that MT neurons integrate retinal image motion with smooth eye movement (*Nadler et al., 2008*; *Nadler et al., 2009*) and global background motion (*Kim et al., 2015a*) signals to compute depth from motion parallax. In addition, most MT neurons are well known to be tuned for binocular disparity (*Maunsell and Van Essen, 1983*; *DeAngelis and Newsome, 1999*; *DeAngelis and Uka, 2003*). Recent studies (*Nadler et al., 2013*; *Kim et al., 2015a*) revealed the existence of many MT neurons that have mismatched depth tuning for motion parallax and binocular disparity cues. Such neurons would presumably not be useful for cue integration in depth perception, and their functional role has thus far remained unclear. In the present study, we demonstrate that such 'opposite' neurons provide valuable signals for detecting object motion during self-motion by selectively responding to local inconsistencies between binocular disparity and motion parallax cues. Thus, our findings provide novel evidence that the functional roles of MT go well beyond representing retinal image motion; they suggest that some MT neurons play fundamental roles in helping to infer the origins, or causes, of retinal image motion.

Our findings have parallels to the potential function of neurons in area MSTd and the ventral intraparietal (VIP) area that have mismatched heading tuning for visual and vestibular cues (*Gu et al., 2006*; *Gu et al., 2008*; *Chen et al., 2011*; *Chen et al., 2013*). Studies of cue integration and cue re-weighting in heading perception have demonstrated that activity of congruent cells can account for behavioral performance (*Gu et al., 2008*; *Fetsch et al., 2011*), but the functional role of opposite cells remained unclear from those studies. More recent work has suggested that opposite neurons may play a role in helping parse the retinal image into signals related to self-motion and object motion (*Kim et al., 2016b*; *Sasaki et al., 2017*), although they did not link opposite cell activity to a relevant behavior. Thus, mismatched tuning, whether unisensory or multisensory, may be a common motif for performing computations that involve parsing sensory signals into components that reflect different causes in the world (*Zhang et al., 2019a*).

More generally, the parsing of retinal image motion into components related to object motion and self-motion is a causal inference problem (*Körding et al., 2007*; *Shams and Beierholm, 2010*; *French and DeAngelis, 2020*), and recent psychophysical work in humans has demonstrated that perception of heading in the presence of object motion follows predictions of a Bayesian causal inference model (*Dokka et al., 2019*). While the neural mechanisms of causal inference are still largely unknown (but see *Fang et al., 2019*), recent computational work has suggested that the relative activity of congruent and opposite cells may provide a critical signal for carrying out causal inference operations (*Zhang et al., 2019b*; *Rideaux et al., 2021*). By providing an empirical link between the activity of opposite cells and detection of object motion during self-motion, our results provide novel evidence for a sensory substrate that may be used to perform causal inference in the domain of object motion and self-motion perception. Elucidating the neural substrates and mechanisms of causal inference regarding object motion is the topic of ongoing studies in our laboratories.

## Materials and methods
### Subjects and surgery

Two male monkeys (*macaca mulatta*, 8–12 kg) participated in these experiments. Standard aseptic surgical procedures under gas anesthesia were performed to implant a head restraint device. A Delrin (Dupont) ring was attached to the skull using a combination of dental acrylic, bone screws, and titanium inverted T-bolts (see *Gu et al., 2006* for details). To monitor eye movements using the magnetic search coil technique, a scleral coil was implanted under the conjunctiva of one eye.

A recording grid made of Delrin was affixed inside the ring using dental acrylic. The grid (2×4×0.5 cm) contains a dense array of holes spaced 0.8 mm apart. Under anesthesia and using sterile technique, small burr holes (~0.5 mm diameter) were drilled vertically through the recording grid to allow the penetration of microelectrodes into the brain via a transdural guide tube. All surgical procedures and experimental protocols were approved by University Committee on Animal Resources at the University of Rochester.

## Experimental apparatus

In each experimental session, animals were seated in a custom-built primate chair that was secured to a six degree-of-freedom motion platform (MOOG 6DOF2000E). The motion platform was used to generate passive body translation along an axis in the fronto-parallel plane, and the trajectory of the platform was controlled in real time at 60 Hz over a dedicated Ethernet link (see *Gu et al., 2006* for details). A field coil frame (C-N-C engineering) was mounted on top of the motion platform to measure eye movements.

Visual stimuli were rear-projected onto a 60×60 cm tangent screen using a stereoscopic projector (Christie Digital Mirage S+3 K) which was also mounted on the motion platform (*Gu et al., 2006*). The display screen was attached to the front side of the field coil frame. To restrict the animal's field of view to visual stimuli displayed on the tangent screen, the sides and top of the field coil frame were covered with matte black enclosures. Viewed from a distance of ~30 cm, the display subtended ~90°×90° of visual angle.

To generate accurate visual simulations of the animal's movement through a virtual environment, an OpenGL camera was placed at the location of one eye, and the camera moved precisely according to the movement trajectory of the platform. Since the motion platform has its own dynamics, we characterized the transfer function of the motion platform, as described previously (*Gu et al., 2006*), and we generated visual stimuli according to the predicted motion of the platform. To account for a delay between the command signal and the actual movement of the platform, we adjusted a delay parameter to synchronize visual motion with platform movement. Synchronization was confirmed by presenting a world-fixed target in the virtual environment and superimposing a small spot by a room-mounted laser pointer while the platform is in motion (*Gu et al., 2006*).

## Electrophysiological recordings

We recorded extracellular single unit activity using single-contact tungsten microelectrodes (FHC Inc) having a typical impedance of 1–3 MΩ. The electrode was loaded into a transdural guide tube and was manipulated with a hydraulic micro-manipulator (Narishige). The voltage signal was amplified and filtered (1 kHz–6 kHz) using conventional hardware (BAK Electronics). Single unit spikes were detected using a window discriminator (BAK Electronics), whose output was time stamped with 1 ms resolution.

Eye position signals were digitized at 1 kHz, then digitally filtered and down sampled to 200 Hz (TEMPO, Reflective Computing). The raw voltage signal from the microelectrode was digitized and recorded to disk at 25 kHz using a Power1401 data acquisition system (Cambridge Electronic Design). If necessary, single units were re-sorted off-line using a template-based method (Spike2, Cambridge Electronic Design).

The location of area MT was initially identified in each animal through analysis of structural MRI scans, which were segmented, flattened, and registered with a standard macaque atlas using CARET software (*Van Essen et al., 2001*). The position of area MT in the posterior bank of the superior temporal sulcus (STS) was then projected onto the horizontal plane, and grid holes around the projection area were explored systematically in mapping experiments. In addition to the MRI scans, the physiological properties of neurons and the patterns of gray matter and white matter encountered along electrode penetrations provided essential evidence for identifying MT. In a typical electrode penetration through the STS that encounters area MT, we first encounter neurons with large RFs and visual motion sensitivity (as expected for area MSTd). This is typically followed by a very quiet region as the electrode passes through the lumen of the STS, and then area MT is the next region of gray matter. As expected from previous studies, RFs of MT neurons are much smaller than those in MSTd (*Komatsu and Wurtz, 1988*) and some MT neurons exhibit strong surround suppression (*DeAngelis and Uka, 2003*) which is typically not seen in MSTd. Confirming a putative localization of the electrode to MT, we observed gradual changes in the preferred direction, preferred disparity, and RF location of multiunit activity, consistent with those described previously (*Albright et al., 1984*; *DeAngelis and Newsome, 1999*).

## Visual stimuli

Visual stimuli were generated by a custom-written C++ program using the OpenGL 3D graphics library (*Kim, 2013*, https://github.com/hkim09/MoogDots_2013) and were displayed using a hardware-accelerated OpenGL graphics card (NVIDIA Quadro FX 1700). The location of the OpenGL

camera was matched to the location of the animal's eye, and images were generated using perspective projection. We calibrated the display such that the virtual environment had the same spatial scale as the physical space through which the platform moved the animal. To view stimuli stereoscopically, animals wore anaglyphic glasses with red and green filters (Kodak Wratten 2 Nos. 29 and 61, respectively). The crosstalk between eyes was measured using a photometer and found to be very small (0.3% for the green filter and 0.1% for the red filter).

## Stimulus to measure depth tuning from motion parallax

We used an established procedure to generate random-dot stimuli to measure depth tuning from motion parallax (*Nadler et al., 2008*). A circular aperture having slightly greater (~10%) diameter than optimal size was located over the center of the RF of the neuron under study. The position of each dot in the image plane was generated by independently choosing random horizontal and vertical locations within the aperture. To present stimuli such that they appear to lie in depth at a specific equivalent disparity, the set of random dots within the circular aperture was ray-traced onto a cylinder corresponding to the desired equivalent disparity, as described in detail previously (*Nadler et al., 2008*). This ray-tracing procedure ensured that the size, location, and density of the random dot patch were constant across simulated depths. Size and occlusion cues were eliminated by rendering transparent dots with a constant retinal size (0.39 deg). Critically, this procedure removed pictorial depth cues and rendered the visual stimulus depth-sign ambiguous, thus requiring interaction of retinal object motion with either extra-retinal signals (*Nadler et al., 2009*) or global visual motion cues (*Kim et al., 2015a*) that specify eye rotation relative to the scene.

The above description assumes lateral translation of the observer in the horizontal plane. In our experiments, animals were translated along an axis in the fronto-parallel plane (i.e. a vertical plane that includes the interaural axis and is parallel to the plane of the display screen) that was aligned with the preferred-null axis of the neuron under study (to elicit robust responses). In this case, we rotated the virtual stimulus cylinder about the naso-occipital axis (normal to the display screen) such that the axis of translation of the observer was always orthogonal to the long axis of the cylinder. This ensures that dots having the same equivalent disparities produce the same retinal speeds regardless of the axis of observer translation (*Nadler et al., 2008*).

## Stimulus for object detection task

Visual stimuli for the main task consisted of a dynamic target object (which could be either moving or stationary in the world), one or three stationary objects (distractors), and a cloud of background dots that appeared outside of a central masked region (*Figure 1—figure supplement 1*, *Video 1*). Background dots were masked out of this central region around the target and distractor objects to avoid having the background dots directly stimulate the RF of the neuron under study. The two-object version of the task (one dynamic target and one stationary distractor) was used in all neural recording experiments, whereas the four-object task (one dynamic target and three stationary distractors) was used during training and in some behavioral control experiments.

For the two-object task, one object was located in the center of the RF of the neuron under study, and the other object was presented on the opposite side of the fixation target (180 deg apart) at the same eccentricity (*Figure 1A, B*). For the four-object task, one object was centered on the RF, and the other three objects were distributed equally (90 deg apart) around the fixation target at equal eccentricities. To present each object at the same retinal position regardless of its depth, the positions of objects were initially determined in screen coordinates and then were ray-traced onto surfaces in the simulated environment (*Figure 1—figure supplement 1B*, left).

Each object was rendered as a square-shaped 'plate' of random dots (density: 1.1 dots/deg$^2$) and was displayed binocularly as a red-green anaglyph. The retinal size of dots was constant (0.15 deg) regardless of object depth, such that dot size was not a depth cue. The target and distractor objects were all of the same retinal size (which was tailored to the RF of the neuron under study) regardless of their location in depth, such that the image size of objects was also not a depth cue. Thus, the only reliable cues to object depth were binocular disparity and motion parallax.

Dynamic target objects had two independent depth parameters, one based on binocular disparity ($d_{BD}$) and the other based on motion parallax ($d_{MP}$). The left-eye and right-eye half-images of the dynamic object were rendered based on the depth defined by binocular disparity, $d_{BD}$. We then

computed the image motion of the dynamic object during translation of the monkey such that it had motion parallax that was consistent with a different depth, $d_{MP}$. Based on the predicted trajectory of the camera on each video frame, we ray-traced the position of the dynamic object (at $d_{MP}$) onto the depth plane defined by binocular disparity, $d_{BD}$ (*Figure 1—figure supplement 1B*, right). This procedure ensures that the dynamic object had a particular difference in depth (ΔDepth, in equivalent disparity units) specified by ($d_{MP} - d_{BD}$), but that it was not possible to detect the dynamic object solely based on its relative motion in the scene (*Figure 1—figure supplement 2*). In other words, when viewed monocularly, the image motion of the dynamic object would be consistent with that of a stationary object at $d_{MP}$. When viewed binocularly, if ΔDepth ≠ 0, the dynamic object's image motion would not be consistent with its depth specified by disparity, $d_{BD}$.

## Experimental protocol

### Preliminary measurements

After isolating the action potential of a single neuron, the RF was explored manually using a small (typically 2–3 deg) patch of random dots. The direction, speed, position, and binocular disparity of the random-dot patch were manipulated using a computer mouse, and instantaneous firing rates were plotted on a display interface that represents the spatial location of the patch in visual space and the stimulus velocity in a direction-speed space. This procedure was used to estimate the location and size of the RF as well as to estimate the neuron's preferences for direction, speed, and binocular disparity.

After these qualitative tests, we measured the direction, speed, binocular disparity, and size tuning of each neuron using quantitative protocols (*DeAngelis and Uka, 2003*). Each of these measurements was performed in a separate block of trials, and each distinct stimulus was repeated 3–5 times. Direction tuning was measured with random dots that moved in eight different directions separated by 45 deg. Speed tuning was measured, at the preferred direction, with random dot stimuli that moved at speeds of 0, 0.5, 1, 2, 4, 8, 16, and 32 deg/s. The stimuli in our main task contained speeds of motion that were <7 deg/s. If a neuron gave very little response (<5 spk/s) to these slow speeds, the neuron was not studied further. Next, the spatial profile of the RF was measured by presenting a patch of random dots at all locations on a 4×4 grid that covered the RF. The height and width of the grid were 1.5–2.5 times larger than the estimated RF size, and each small patch was approximately ¼ the size of the RF. Responses were fitted by a 2D Gaussian function to estimate the center location and size of the RF. To measure binocular disparity tuning, a random dot stereogram was presented at binocular disparities ranging from –2 deg to +2 deg in steps of 0.5 deg. For this disparity tuning measurement, dots moved in the neuron's preferred direction and speed. Finally, size tuning was measured with random-dot patches having diameters of 0.5, 1, 2, 4, 8, 16, and 32 deg.

Depth tuning from motion parallax was then measured as described previously (*Nadler et al., 2008*; *Nadler et al., 2013*). Dots were presented monocularly and were rendered at one of nine simulated depths based on their motion (–2 deg to +2 deg of equivalent disparity in steps of 0.5 deg), in addition to the null condition in which only the fixation target was presented. Each distinct stimulus was repeated 6–10 times. During measurement of depth tuning from motion parallax, animals underwent passive whole-body translation which followed a modified sinusoidal trajectory along an axis in the fronto-parallel plane (*Figure 1—figure supplement 1C*). To smooth the onset and offset, the 2 s sinusoidal trajectory was multiplied by a Gaussian function that was exponentiated to a large power as follows:

$$G\left(t\right) = e^{-\frac{(t-t_0)^n}{\sigma^n}}$$

where $t_0$=1.0 s, σ=0.92, and n=22. On half of the trials, platform movement started toward the neuron's preferred direction. On the other half, motion started toward the neuron's null direction (*Figure 1—figure supplement 1C*). During body translation, animals were required to maintain fixation on a world-fixed target, which required a compensatory smooth eye movement in the direction opposite to head movement.

### Moving object detection task

We presented one dynamic (i.e. moving) object and one (or three) stationary object(s) while the animal experienced the modified sinusoidal lateral motion as described above. The animal was trained to

identify the dynamic object by making a saccadic eye movement to it (*Figure 1A*). At the beginning of each trial, the fixation target first appeared at the center of the screen. After the animal established fixation for 0.2 s, the dynamic object, stationary object(s), and background cloud of dots appeared and began to move as the animal was translated sinusoidally for 2.1 s (see *Video 1*). Because the fixation target was world-fixed, translation of the animal required a counter-active smooth eye movement to maintain visual fixation. An electronic window around the fixation target was used to monitor and enforce pursuit accuracy. The initial size of the target window was 3–4 deg, and it shrank to 2.1–2.8 deg after 250 ms of translation. This allowed the animal a brief period of time to initiate pursuit and execute a catch-up saccade to arrive on target. At the end of visual stimulation, both the fixation target and the visual stimuli disappeared and a choice target (0.4 deg in diameter) appeared at the center location of each object. The animal then attempted to make a saccadic eye movement to the location of the dynamic object and received a liquid reward (0.2–0.4 ml) for correct answers.

Based on the preliminary tests described above, we set the axis of translation within the fronto-parallel plane to align with the preferred-null axis of the neuron under study. In the main detection task, we systematically varied the depth discrepancy (ΔDepth) between disparity and motion parallax cues for the dynamic object to manipulate task difficulty (*Figure 1—figure supplement 1B*). ΔDepth is defined as the difference between depths specified by motion parallax and binocular disparity cues, ($d_{MP} - d_{BD}$). Different values of ΔDepth were applied to the dynamic object around a fixed 'pedestal depth' (red line, *Figure 1—figure supplement 1B*). For the vast majority of recording sessions, the pedestal depth was fixed at –0.45 deg (103/106 sessions), although it deviated from this value slightly in a few early experiments. We elected to use a fixed pedestal depth such that all neurons were tested with the same stimulus values, thereby allowing for decoding analyses (described below). The pedestal depth was chosen as the average midpoint between the preferred depths obtained from tuning curves for disparity and motion parallax, based on data from a previous study (*Nadler et al., 2013*). We used the following ΔDepth values: –1.53, –0.57, –0.21, 0, 0.21, 0.57, and 1.53 deg. Stationary objects were presented at one of seven possible depths (–1.6 deg to +1.6 deg in steps of 0.4 deg). The vast majority of recording sessions were conducted using these 'standard' pedestal depth, ΔDepth, and stationary depth values (101/106 sessions). Thus, the maximum range of depths of dynamic objects (–1.215 to +0.315 deg) was well within the range of depths for stationary objects, which ensured that the animals could not perform the task solely based on depth outliers (either in binocular disparity or motion parallax). The identity of each object (dynamic/stationary) and its depth values were chosen from the above ranges randomly on each trial. Each ΔDepth value of the dynamic object was repeated at least 14 times (mean: 35 and SD: 9.6).

For three sessions, a monkey performed the object detection task without binocular disparity cues in a fraction of trials (*Figure 1—figure supplement 2*, monocular condition). In this control condition, the visual stimulus (except for the fixation point) was displayed to only one eye in 16% of trials, while the rest of the task structure remained the same. Monocular conditions were presented in a small percentage of trials in order not to frustrate the animal, given that performance was poor on these monocular trials.

## Animal training procedure

Although the object detection task is conceptually simple, it required extensive behavioral training, involving a number of steps. Here, we outline a series of operant conditioning steps required to teach animals to perform the task. Following basic chair training and habituation to the laboratory, animals were trained to maintain visual fixation on a target during sinusoidal translation of the motion platform.

Once smooth eye movements tracked the fixation target with pursuit gains approaching 0.9, we initially trained animals to detect a moving object without any self-motion, such that any motion of an object on the display resulted from object motion relative to the scene. After fixation, four objects appeared on the display and only one of them moved sinusoidally along a horizontal trajectory for 2.1 s. In the early stages of this training, a saccade target appeared only at the location of the moving object. Subsequently, we introduced a fraction of trials in which saccade targets appeared at the locations of all four objects, and we gradually increased the proportion of these trials. During this phase of training, the depths of the objects, as defined solely by binocular disparity since there

was no self-motion, were randomly drawn from a uniform distribution spanning the range from –1.6 to +1.6 deg, to help animals generalize the task.

Once animals performed the task well in the absence of self-motion, we began to introduce small amounts of sinusoidal self-motion, which induced subtle retinal image motion of all objects. During the initial stages of this training period, the dynamic object had a large motion amplitude such that it was quite salient relative to the motion of stationary objects that was due to self-motion. As the animals became accustomed to performing the task during self-motion, we gradually increased the magnitude of self-motion (up to 2.8 cm) and decreased the motion amplitude of the dynamic object. Once the retinal motion amplitude of the dynamic object became comparable to that of stationary objects, we began to introduce a depth discrepancy between disparity and motion parallax (ΔDepth). That is, the motion trajectory of the dynamic object began to follow that of an object at a different depth, $d_{MP}$ (*Figure 1—figure supplement 1B*, right). We used a staircase procedure to train animals over a range of values of ΔDepth. During this phase of training, we interleaved three different pedestal depths (–0.51 deg, 0 deg, and 0.51 deg) to help animals generalize the task, and we randomly chose the depths of the three stationary objects from the range –1.6 to +1.6 deg.

Once we observed stable 'v-shaped' psychometric functions for all three pedestal depths over a span of more than 10 days (e.g. *Figure 1—figure supplement 3A, B*), we transitioned to the final stimulus configuration for recording experiments. To keep the number of stimulus conditions manageable for recording, this configuration included one pedestal depth and two objects (one dynamic and one stationary). Following recording experiments, we revisited the more general version of the task involving four objects and three pedestal depths to make sure that behavioral performance did not reflect any change in strategy (e.g. *Figure 1—figure supplement 3C*).

## Data analyses

### Regression analysis of behavior

We used multinomial regression to assess the relative contributions of $d_{BD}$, $d_{MP}$, and ΔDepth to perceptual decisions. If animals perform the task primarily based on the discrepancy between disparity and motion parallax cues to depth, we expect to see a much greater contribution of ΔDepth relative to $d_{BD}$ and $d_{MP}$. For each possible choice location, i, (i.e. a chosen location or a not-chosen location), we performed the following regression:

$$\log \left( \frac{P(choice_i)}{1-P(choice_i)} \right) = \beta_0 + \sum_{j}^{N} \left( \beta_{BD,i,j} \left| d_{BD,i,j} \right| + \beta_{MP,i,j} \left| d_{MP,i,j} \right| + \beta_{\Delta,i,j} \left| \Delta\text{Depth}_{i,j} \right| \right) \tag{1}$$

where j denotes the locations of objects on the screen, and N is the total number of objects (two or four). Once beta values were obtained, we averaged betas across the two (or four) possible choice locations and also averaged betas across the two (or 12) not chosen locations (*Figure 1D*, *Figure 1—figure supplement 3D, E*).

We also quantified the proportion of fits that produced significant values of each beta coefficient (*Figure 1E*). The number of beta values significantly different from zero (alpha = 0.05) were summed across locations (two or four) and across sessions. The results were then divided by the total number of beta values (2 * number of valid sessions or 4 * number of valid sessions, respectively). For not-chosen objects in the four-object task, the number of significant fits were summed across three locations and then divided by 12 * number of valid sessions.

### Depth-sign tuning and discrimination index

Average firing rates during stimulus presentation were plotted as a function of simulated depth (*Figure 2A–C*) to construct depth tuning curves. To quantify the relative strength of neural responses to near and far depths defined by binocular disparity or motion parallax, we computed a DSDI from each tuning curve (*Nadler et al., 2008*; *Nadler et al., 2009*).

$$DSDI = \frac{1}{4} \sum_{i=1}^{4} \frac{R_{far(i)} - R_{near(i)}}{\left| R_{far(i)} - R_{near(i)} \right| + \sigma_{avg(i)}} \tag{2}$$

For each pair of depths symmetrical around zero (for example, ±2 deg), the difference in mean response between far ($R_{far}$) and near ($R_{near}$) depths was computed relative to response variability ($\sigma_{avg}$,

the average SD of responses to the two depths). This quantity was then averaged across the four pairs of depth magnitudes to obtain the DSDI ($-1<DSDI<+1$). Near-preferring neurons have negative DSDI values, whereas far-preferring neurons have positive DSDI values. Statistical significance of DSDI values was evaluated using a permutation test in which DSDI values were computed 1000 times after shuffling responses across depths. If the measured DSDI value is negative, the p value is the proportion of shuffled DSDIs less than the measured DSDI value. If the measured DSDI is positive, the p value is the proportion of DSDIs greater than the measured DSDI value.

### Depth sign discrimination index for dynamic object tuning

Average firing rates during stimulus presentation were plotted as a function of depth difference (*Figure 2D–F*) to construct dynamic object tuning curves. To quantify the relative strength of neural responses to negative and positive values of ΔDepth, we computed a DSDI metric for the dynamic object responses ($DSDI_{dyn}$) as follows:

$$DSDI_{dyn} = \frac{1}{3} \sum_{i=1}^{3} \frac{R_{pos(i)} - R_{neg(i)}}{\left| R_{pos(i)} - R_{neg(i)} \right| + \sigma_{avg(i)}} \tag{3}$$

For each pair of ΔDepth values symmetrical around zero (e.g. ±1.53 deg), the difference in mean response between positive ($R_{pos}$) and negative ($R_{neg}$) ΔDepth was computed relative to response variability ($\sigma_{avg}$, the average SD of responses to the two ΔDepth values). This quantity was then averaged across the three pairs of ΔDepth values to obtain $DSDI_{dyn}$ ($-1<DSDI_{dyn}<+1$).

### Depth tuning congruency

Congruency of depth tuning curves obtained by manipulating binocular disparity and motion parallax cues was quantified using a correlation coefficient. The Pearson correlation was computed between the two cues using the average responses across nine depths (−2 to 2 deg in steps of 0.5 deg) for each cue; this coefficient is noted as $R_{MP\_BD}$ (*Figure 3B*). Neurons were classified as 'congruent' or 'opposite' if their value of $R_{MP\_BD}$ was significantly greater or less than zero, respectively.

### Neurometric performance

We used an ideal observer analysis to measure how reliably single neurons can signal whether an object is dynamic or stationary. For each value of ΔDepth, the distribution of firing rates across trials was sorted into two groups according to the type of object in the RF (dynamic vs. stationary). An ROC curve was computed from the pair of response distributions for each ΔDepth (*Britten et al., 1992*), and performance of the ideal observer was defined as the area under the ROC curve. ROC areas were then plotted as a function of ΔDepth to construct a neurometric function (*Figure 5B, E*). To obtain a single measure of NP, we then averaged the ROC areas across non-zero values of ΔDepth to obtain a single metric for each neuron. This average ROC area will be >0.5 if a neuron responds preferentially to dynamic objects overall and <0.5 if it responds preferentially to stationary objects overall.

### Detection probability

DP is a measure of the relationship between neural responses and perceptual decisions in a detection task (*Bosking and Maunsell, 2011*) and is similar to the choice probability metric (*Britten et al., 1996*). The procedure for computing DP is analogous to the ROC analysis described above, except that responses are sorted into two groups according to the animal's perceptual decision (dynamic vs. stationary object in the RF). To eliminate any contamination from stimulus effects, only ambiguous trials (ΔDepth = 0) were used to compute DP (*Figure 4*). A permutation test was used to determine whether each DP value was significantly different from the chance level of 0.5 (*Uka and DeAngelis, 2004*).

### Decoding analyses

We constructed an optimal linear decoder to detect moving objects based on simulated responses from a population of 97 model neurons. Model neurons correspond to the dominant subset of recorded neurons for which data were collected under identical stimulus conditions, thus allowing us to construct pseudo-population responses. We randomly selected 100,000 samples of stimulus

conditions from the datasets with replacement (16 unique stimulus conditions within the RF). The mean and SD of measured responses to each stimulus condition were then used to generate simulated responses according to the following equation (*Shadlen et al., 1996*; *Cohen and Newsome, 2009*; *Gu et al., 2014*):

$$Response = \ \mu + Q \times r_{rand} \times \sigma \tag{4}$$

where $\mu$ and $\sigma$ are vectors of means and SDs of the population across stimulus conditions, $r_{rand}$ is a vector of standard normal deviates (MATLAB 'normrnd' function with zero mean and unity standard deviation), and Q is the square root of the correlation matrix. The correlation matrix was modeled such that pairs of neurons with similar NP values have stronger correlated noise, and pairs of neurons with dissimilar NP values show weaker correlated noise:

$$r\_noise_{i,j} = 1.1 \ \times \left( 0.5 - \sqrt{\left| NP_i - NP_j \right|} \right) \tag{5}$$

where $NP_i$ is the neurometric performance of neuron i. This generated noise correlations (0.15±0.17, mean ± SD) of roughly similar strength to those observed in empirical studies of MT neurons (*Zohary et al., 1994*; *Huang and Lisberger, 2009*).

Total trials were divided into training (90%) and test (10%) sets. A linear decoder was trained to classify whether the stimulus in the RF was a dynamic or stationary object based on population responses in the training set. We used linear discriminant analysis (MATLAB 'classify' function) to determine the weights of the decoder. Ambiguous trials (ΔDepth = 0) were excluded from the training set.

The test set was used to validate performance of the decoder. A DP_pred was computed for each neuron in the model in the same way we computed DP from the empirical data, except that the decoder's 'choice' for each trial was used instead of the monkey's behavioral choice. Specifically, responses to ambiguous stimuli (ΔDepth = 0) in the test set were sorted according to the decoder's output (dynamic vs. stationary object prediction).

## Time course of choice-related responses

Spikes in the ambiguous trials (ΔDepth = 0) were aligned to stimulus onset, compiled into peri-stimulus time histograms, and then smoothed using a 150 ms boxcar window. Trials were first sorted by the phase of self-motion (phase 0 or phase 180), and then sorted by the animal's choice (whether the animal chose an object within the RF or not). Average responses were z-scored using a session-wide mean and SD. We plotted the mean and SE of the z-scored responses, as well as the difference in z-scored responses between choices (*Figure 4—figure supplement 1*). For each phase, we tested whether the median responses for the two choices at each time point were significantly different or not (alpha = 0.05, Wilcoxon signed-rank test).

## Neuron samples and selection criteria

We analyzed data from a total of 123 single units (53 neurons from M1 across 73 recording sessions and 70 neurons from M2 across 82 recording sessions) for which we completed the basic tuning measurements, including tuning for direction, speed, RF position, size, depth from binocular disparity, and depth from motion parallax. Among these, we completed the object detection task for 106 neurons (47 from M1 and 59 from M2). This set of 106 neurons constitutes the sample for the single neuron analyses of *Figure 3*. Except for two neurons, 104 of these 106 neurons were tested using a standard set of ΔDepth values, including zero (47 from M1 and 57 from M2).

To compute detection probability, we analyzed a subset of these 104 neurons for which the monkey made at least five choices in favor of both target locations when ΔDepth = 0 (92 neurons, 39 from M1 and 53 from M2). For population decoding (*Figure 6*), we required that each dataset contain responses to objects at all of the standard depth values for the stationary object. Three neurons were excluded because they were tested with slightly different stationary depth values, and four neurons were excluded because they did not have responses to stationary objects at all of the standard depth values (which can occur because the depths of stationary objects were chosen randomly from the standard values in each trial). Thus, with these exclusions, 97 neurons contributed to the population decoding analysis (45 from M1 and 52 from M2).

## Acknowledgements

We thank Johnny Wen for programming assistance, as well as Swati Shimpi, Emily Murphy, and Dina Graf for assistance with training animals. This work was supported by NEI R01 grant EY013644, NINDS U19 grant NS118246, and by an NEI Core grant (EY001319).

## Additional information

### Funding

| Funder | Grant reference number | Author |
| --- | --- | --- |
| National Eye Institute | EY013644 | Gregory C DeAngelis |
| National Institute of Neurological Disorders and Stroke | NS118246 | Dora E Angelaki<br>Gregory C DeAngelis |
| National Eye Institute | EY001319 | Gregory C DeAngelis |

The funders had no role in study design, data collection and interpretation, or the decision to submit the work for publication.

### Author contributions

HyungGoo R Kim, Conceptualization, Data curation, Formal analysis, Investigation, Methodology, Software, Visualization, Writing – original draft, Writing – review and editing; Dora E Angelaki, Conceptualization, Writing – review and editing; Gregory C DeAngelis, Conceptualization, Funding acquisition, Project administration, Supervision, Validation, Writing – original draft, Writing – review and editing

### Author ORCIDs

HyungGoo R Kim http://orcid.org/0000-0002-9106-4960
Dora E Angelaki http://orcid.org/0000-0002-9650-8962
Gregory C DeAngelis http://orcid.org/0000-0002-1635-1273

### Ethics

All surgical procedures and experimental protocols were approved by the University Committee on Animal Resources at the University of Rochester (#100682) and were performed in accordance with the recommendations in the Guide for the Care and Use of Laboratory Animals of the National Institutes of Health.

### Decision letter and Author response

Decision letter https://doi.org/10.7554/eLife.74971.sa1
Author response https://doi.org/10.7554/eLife.74971.sa2

## Additional files

### Supplementary files

• Transparent reporting form

### Data availability

Data have been made available on Figshare: https://doi.org/10.6084/m9.figshare.19783624.

The following dataset was generated:

| Author(s) | Year | Dataset title | Dataset URL | Database and Identifier |
| --- | --- | --- | --- | --- |
| HyungGoo K, Angelaki ED, DeAngelis GC | 2022 | A neural mechanism for detecting object motion during self-motion | https://doi.org/10.6084/m9.figshare.19783624 | figshare, 10.6084/m9.figshare.19783624 |

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
