## [Editor Report]

This paper will be of broad interest to readers in the field of visual processing. The authors use concurrent psychophysics and single unit recordings, along with modeling, to investigate how visual signals in primate cortical area MT can distinguish between visual motion induced by self-motion and the motion of other objects in the world. The experiments provide an explanation for otherwise puzzling discrepancies in the depth tuning of MT cells.

---

## [Decision Letter]

**Decision letter after peer review:**

Thank you for submitting your article "A neural mechanism for detecting object motion during self-motion" for consideration by *eLife*. Your article has been reviewed by 3 peer reviewers, one of whom is a member of our Board of Reviewing Editors, and the evaluation has been overseen by Tirin Moore as the Senior Editor. The following individuals involved in review of your submission have agreed to reveal their identity: Anirvan S Nandy (Reviewer #2); Oliver W. Layton (Reviewer #3).

Essential revisions:

The reviewers agreed that this was an interesting study and with careful controls. During the discussion, there were 3 consensus points that will require addressing in your revision, as you will also see in the individual reviews below.

1) There should be more discussion of the relationship between the incongruency mechanism explored here and the more general mechanism of flow parsing. How would this mechanism relate to or be combined with or substitute for flow parsing under different conditions?

2) The authors should perform neurometric analyses and modeling to more directly substantiate their claims. In particular, the population of cells included in this analysis could be more directly tied to populations of cells that include no incongruent cells, include only incongruent cells, or include some range of congruencies, to see directly how these different tunings in a population of cells affect decoding. This would go further than the current less direct analyses, and strengthen conclusions about the role of incongruent cells.

3) In Figure S2, two reviewers were a bit confused about the implications of a monkey getting the monocular presentation wrong 90-95% of the time, if we are reading that plot correctly. Reviewers expected this to be more like 50% performance. How should this be interpreted?

*Reviewer #1 (Recommendations for the authors):*

1) In this paper, starting in the abstract, the authors emphasize the role played by these incongruent MP/BD neurons. In the discussion, they say (quoting in full because there are no line numbers to reference): "our findings support the idea that a sensory representation consisting of a mixture of congruent and opposite cells provides a useful sensory substrate for causal inference (Rideaux et al. 2021). To our knowledge, these findings provide the first empirical evidence for a specific contribution of opposite neurons to perceptual inference about causes of sensory signals." However, as far as I could tell, the analyses and modeling never actually pulled out the incongruent neurons in particular to analyze them in isolation or analyze the performance without them. Thus, I had a hard time feeling this claim was supported.

In particular, in Figure 5G, the authors conclude: "Thus, the MT neurons that most strongly predict decisions to detect the dynamic object (on ambiguous trials) are those with incongruent tuning that makes them selective for dynamic objects." But this is a bit indirect, because the figure shows correlation between detection probability and neurometric performance, and the neurometric performance is correlated with congruency. Why not just correlate DP with a direct measure of non-congruence?

(On that same plot (Figure 5G), I'm puzzled why there are so few DP and NP values <0.5, when roughly half of all neurons were congruent in their BD and MP tuning – wouldn't one expect those neurons to have NP<0.5 and DP<0.5?)

Most generally, I'm still unclear about how much congruent neurons can contribute to these decisions compared to the incongruent ones. Given how well the population decoding works, it would be interesting to know if it still worked this well with *only* congruent or *only* incongruent neurons included. Is there an alternative hypothesis where having a range of different MP vs. BD tunings is what allows correct identification of the moving object, and it's the tuning variability rather than the incongruent neurons per se that matters? One could test this directly by including different ranges of neuronal tunings into a predictive model. Overall, it seems like there should be more direct analyses of the incongruent neuron population to provide support for the strong claims the authors are making about their role.

2) Figure S2: I'm a little puzzled at how a monkey could get it wrong 95% of the time in the monocular +1.5 degree \Δ Depth condition. What's going on there? Doesn't this mean that there *are* monocular cues for solving this task and they're just being interpreted entirely wrongly by the monkey? How should this be interpreted?

*Reviewer #2 (Recommendations for the authors):*

I would like the authors to consider these two points:

(A) In each experimental session, the axis of translation was aligned with the preferred axis of the neuron under observation in order to elicit robust responses. What would be the role of these incongruent neurons when self-motion is not aligned to this axis? Will it be possible to show that a population code of such neurons is sufficient to detect dynamic objects in the face of an arbitrary axis of self-motion?

(B) The authors state that it is not a worthwhile exercise to find model parameters that best match the empirical data, noting that they would have to make assumptions about the structure of correlated noise. It appears that in their dataset they should have several sets of simultaneously recorded neurons (e.g. 70 neurons from 57 sessions in M2). Might these give a hint to this correlation structure?

*Reviewer #3 (Recommendations for the authors):*

I have a few questions and some suggestions that I think would improve the manuscript and better contextualize the findings.

1. The stimulus in the present study contains background dots in the periphery outside the masked region that produce retinal motion consistent with the observer's self-motion. It would be helpful if the authors clarified the extent to which this peripheral motion may activate MSTd neurons and potentially recruit a flow parsing mechanism. In other words, is it possible that the background motion engages flow parsing, perhaps interacting with the proposed mechanism to some extent?

2. In several places in the paper (e.g. bottom of p.3, p. 26, and p. 28) the authors' descriptions may be taken to mean that the proposed local mechanism circumvents the need for flow parsing. Some examples: "it allows detection of object motion without the need for more complex computations that discount the global flow field" (p.3); "this mechanism… may be relatively economical for the nervous system to implement" (p.3); " An advantage of our proposed mechanism over flow parsing is that it does not require estimation of the global flow field, nor a complicated mechanism …" (p. 28). Given that the authors acknowledge elsewhere in the manuscript that the proposed mechanism likely complements flow parsing, I think the authors would agree that the proposed mechanism is not a direct substitute in general and rather is likely synergistic (e.g. multiple solutions to the same problem enhance robustness in different contexts, the proposed mechanism may be especially useful when quick reaction time is needed, etc.). Flow parsing is a much broader process and addresses far more than moving object detection. I would suggest rephrasing some of these statements and being cautious with language like "advantages over" flow parsing because the proposed local mechanism is more specialized than flow parsing.

3. I think the limitations from the public review should be referenced/discussed in the paper.

4. Warren and Rushton (2009) have demonstrated that humans parse object motion from self-motion to a similar extent regardless of whether the background motion surrounding a moving object is present in a monocular display. Niehorster and Li (2017) found a similar result for stereo optic flow. Simulations by Layton and Fajen (2016; 2020) and Layton and Niehorster (2019) support the hypothesis that a global mechanism plays a primary role. Together, these findings suggest that local mechanisms may play a much smaller role, at least for flow parsing. How do these findings relate to the proposed mechanism?

5. I found Supplementary Figure 1b very helpful to develop an intuition about the displays (as was the included video). I know figure space is at a premium, but if possible, including Supplementary Figure 1b in the main manuscript would likely help readers because the stimulus design is rather complex.

6. Aspects of the paragraph related to opposite at the bottom of p. 29 sounded repetitive after reading the previous parts of the paper and I thought they could be condensed/removed.

7. I found the motion parallax light blue color difficult to see in Figure 2. I would appreciate it if the authors replaced it with a more salient color.

8 Supplementary Figure 2: Why is the proportion correct so much less than 50% on the monocular task? Given that the monocular task is very challenging and there are two possibilities, I would have expected the proportion correct to be roughly 50%.

9. p. 7. I don't think I understand the following statement and would appreciate clarification: "…animals were translated along an axis in the fronto-parallel plane that was aligned with the preferred-null axis of the neuron under study." Does fronto-parallel plane mean rotation about the Z axis (axis toward the stimulus)?

---

## [Author Response]

Essential revisions:The reviewers agreed that this was an interesting study and with careful controls. During the discussion, there were 3 consensus points that will require addressing in your revision, as you will also see in the individual reviews below.1) There should be more discussion of the relationship between the incongruency mechanism explored here and the more general mechanism of flow parsing. How would this mechanism relate to or be combined with or substitute for flow parsing under different conditions?2) The authors should perform neurometric analyses and modeling to more directly substantiate their claims. In particular, the population of cells included in this analysis could be more directly tied to populations of cells that include no incongruent cells, include only incongruent cells, or include some range of congruencies, to see directly how these different tunings in a population of cells affect decoding. This would go further than the current less direct analyses, and strengthen conclusions about the role of incongruent cells.3) In Figure S2, two reviewers were a bit confused about the implications of a monkey getting the monocular presentation wrong 90-95% of the time, if we are reading that plot correctly. Reviewers expected this to be more like 50% performance. How should this be interpreted?

We have addressed and resolved all three of these issues. Here is a brief summary of how the consensus points have been addressed, with further details described below. (1) We have added considerable discussion of the relationship between this new mechanism and flow parsing. We agree with Reviewer #3 that they are complementary mechanisms. (2) We have added new single-cell-level analyses of the relationships between incongruency and both detection probability and neurometric performance (new Figure 5—figure supplement 1). We have also now separately decoded incongruent and congruent neurons and demonstrate better performance by incongruent neurons (new Figure 7). (3) We have cleared up the issue of the behavioral control performance. Performance was less 50% because these data came from the 4-alternative version of the task (25% chance level). In addition to addressing these consensus issues, we have addressed all of the other specific points of the reviewers, as detailed below.

Reviewer #1 (Recommendations for the authors):1) In this paper, starting in the abstract, the authors emphasize the role played by these incongruent MP/BD neurons. In the discussion, they say (quoting in full because there are no line numbers to reference): "our findings support the idea that a sensory representation consisting of a mixture of congruent and opposite cells provides a useful sensory substrate for causal inference (Rideaux et al. 2021). To our knowledge, these findings provide the first empirical evidence for a specific contribution of opposite neurons to perceptual inference about causes of sensory signals." However, as far as I could tell, the analyses and modeling never actually pulled out the incongruent neurons in particular to analyze them in isolation or analyze the performance without them. Thus, I had a hard time feeling this claim was supported.In particular, in Figure 5G, the authors conclude: "Thus, the MT neurons that most strongly predict decisions to detect the dynamic object (on ambiguous trials) are those with incongruent tuning that makes them selective for dynamic objects." But this is a bit indirect, because the figure shows correlation between detection probability and neurometric performance, and the neurometric performance is correlated with congruency. Why not just correlate DP with a direct measure of non-congruence?

These are very reasonable criticisms, and have led us to explore the relationships between incongruency, neurometric performance, and detection probability further. First, we now show (Figure 5—figure supplement 1A) that neurometric performance is significantly correlated with peak response ratio. This result shows that neurons with peak response ratios substantially greater than unity almost always have neurometric performance > 0.5. However, this result also shows that there are neurons with peak response ratios near unity (including slightly less than unity) that also have high neurometric performance (such as the example neuron in Figure 2C,F).

On the reviewer’s suggestion, we also looked at the relationship between DP and peak response ratio directly, as now shown in Figure 5—figure supplement 1B. Although there is a trend in the expected direction, it is not significant. The main reason for this appears to be that there is a cluster of neurons with large peak response ratios that have DP values near 0.5. Digging into this further, we discovered that these neurons generally have tuning properties similar to the example cell in Figure 2B,E. In cases like this, while incongruent tuning creates a clear preference for dynamic objects, it is limited to a narrow range of depth values, and some dynamic objects elicit responses that are weaker than many stationary objects. Thus, what seems to be crucial for producing high DP values is that a neuron *consistently* prefers dynamic objects over the range tested. Thus, the monkey may have adopted a strategy of selectively reading out neurons that consistently prefer dynamic objects across the stimulus range. In this case, neurons with high peak response ratios but strongly varying responses to dynamic objects (Figure 2 B,E) do not appear to be well correlated with decisions.

We have now expanded the Results section (pp. 11-12) to present and discuss these findings. We have also modified text throughout the manuscript to reflect our finding that correlation with decisions depends mainly on a consistent preference for dynamic objects rather than the presence of incongruency per se.

(On that same plot (Figure 5G), I'm puzzled why there are so few DP and NP values <0.5, when roughly half of all neurons were congruent in their BD and MP tuning – wouldn't one expect those neurons to have NP<0.5 and DP<0.5?)

We think there are two parts of an answer to this question. First, the reviewer seems to assume that if NP < 0.5 then DP should also be <0.5. But this doesn’t necessarily need to be the case. For neurons that prefer stationary objects, it is certainly possible that they are mainly uncorrelated with decisions rather than being anticorrelated. Second, the reviewer asks why NP values are not symmetrically distributed around 0.5 in Figure 5G. This is a good question; our analyses suggest that this arises because of the distribution of our stimulus values for the dynamic object condition. Because almost all neurons were tested with a pedestal depth of -0.45 deg, the depth values for both disparity and motion parallax tend to be mostly negative for the dynamic object conditions (this can be observed by looking at the two x-axes in Figure 2D-F). This bias toward negative (near) values of dynamic objects, combined with the fact that most neurons have a near preference for depth from motion parallax (Figure 3A), means that most neurons, including most congruent cells, tended to have mean responses to dynamic objects that were above the mean response to stationary objects (e.g., Figure 2D). And this is captured by the neurometric performance. We now discuss this asymmetry in the text on p. 11.

Most generally, I'm still unclear about how much congruent neurons can contribute to these decisions compared to the incongruent ones. Given how well the population decoding works, it would be interesting to know if it still worked this well with *only* congruent or *only* incongruent neurons included. Is there an alternative hypothesis where having a range of different MP vs. BD tunings is what allows correct identification of the moving object, and it's the tuning variability rather than the incongruent neurons per se that matters? One could test this directly by including different ranges of neuronal tunings into a predictive model. Overall, it seems like there should be more direct analyses of the incongruent neuron population to provide support for the strong claims the authors are making about their role.

We thank the reviewer for making this suggestion. We have now performed decoding simulations after breaking up the population into three groups based on the peak response ratio measure of preference for dynamic objects. These new results show that neurons with high peak response ratios produce better decoding performance and greater predicted detection probabilities than neurons with low peak response ratios. These results are shown in a new Figure 7 and described in the associated text on p. 14. We also found similar results when dividing the population into groups based on the correlation between depth tuning for disparity and motion parallax R_MP_BD_ (see text). We think that these new findings address the reviewer’s concern by demonstrating directly that neurons with incongruent tuning enable better performance for decoding scene-relative object motion.

2) Figure S2: I'm a little puzzled at how a monkey could get it wrong 95% of the time in the monocular +1.5 degree \Δ Depth condition. What's going on there? Doesn't this mean that there *are* monocular cues for solving this task and they're just being interpreted entirely wrongly by the monkey? How should this be interpreted?

We thank Reviewers #1 and #3 for questioning this. It turns out that these control data were from the version of the task with 4 objects, rather than the version with two objects. Thus, the chance level of performance (now indicated in the figure) is 25%. This accounts for most of the mystery regarding the monocular condition. The remaining variation around the chance level is presumably just somewhat random as the monkey was only exposed to the monocular condition in 16% of trials across 3 sessions. The figure and caption have been modified to clarify this issue (now Figure 1—figure supplement 2).

Reviewer #2 (Recommendations for the authors):I would like the authors to consider these two points:(A) In each experimental session, the axis of translation was aligned with the preferred axis of the neuron under observation in order to elicit robust responses. What would be the role of these incongruent neurons when self-motion is not aligned to this axis? Will it be possible to show that a population code of such neurons is sufficient to detect dynamic objects in the face of an arbitrary axis of self-motion?

We don’t have any data to address this question directly because we did not record from neurons while we intentionally misaligned the stimulus axis with the preferred direction. That said, MT neurons are rather broadly tuned, and so we would expect their response modulations to fall off gradually with such a misalignment. In this sense, we would not expect this case to be substantially different from any analogous situation involving a population of neurons with broadly distributed preferences. As the axis of motion changes, some neurons will become less informative while others will become more informative.

(B) The authors state that it is not a worthwhile exercise to find model parameters that best match the empirical data, noting that they would have to make assumptions about the structure of correlated noise. It appears that in their dataset they should have several sets of simultaneously recorded neurons (e.g. 70 neurons from 57 sessions in M2). Might these give a hint to this correlation structure?

Unfortunately, there seems to be a misunderstanding here. There were not 70 neurons recorded from 57 sessions in M2 during the discrimination task. In fact, we didn’t have any well-isolated cell pairs that were recorded through the main detection task. For M2, the 70 single units that were recorded during the preliminary tests came from 82 recording sessions (as now clarified in Methods, p. 34, for both animals). Thus, we don’t have sufficient data to make any reasonable attempt at constraining the correlation structure.

Reviewer #3 (Recommendations for the authors):I have a few questions and some suggestions that I think would improve the manuscript and better contextualize the findings.1. The stimulus in the present study contains background dots in the periphery outside the masked region that produce retinal motion consistent with the observer's self-motion. It would be helpful if the authors clarified the extent to which this peripheral motion may activate MSTd neurons and potentially recruit a flow parsing mechanism. In other words, is it possible that the background motion engages flow parsing, perhaps interacting with the proposed mechanism to some extent?

We have not recorded from MSTd neurons using these stimuli, so we do not have a direct answer to this question based on data. However, we think it is reasonable to assume that the background motion would drive at least some MSTd neurons. Thus, it is possible that flow parsing could also be engaged to some extent in our task, as now acknowledged explicitly on p. 17. However, as the reviewer notes in a previous comment, flow parsing does not require disparity signals whereas the local mechanism demonstrated by our findings does. Hence, it seems unlikely that flow parsing plays a major role here.

2. In several places in the paper (e.g. bottom of p.3, p. 26, and p. 28) the authors' descriptions may be taken to mean that the proposed local mechanism circumvents the need for flow parsing. Some examples: "it allows detection of object motion without the need for more complex computations that discount the global flow field" (p.3); "this mechanism… may be relatively economical for the nervous system to implement" (p.3); " An advantage of our proposed mechanism over flow parsing is that it does not require estimation of the global flow field, nor a complicated mechanism …" (p. 28). Given that the authors acknowledge elsewhere in the manuscript that the proposed mechanism likely complements flow parsing, I think the authors would agree that the proposed mechanism is not a direct substitute in general and rather is likely synergistic (e.g. multiple solutions to the same problem enhance robustness in different contexts, the proposed mechanism may be especially useful when quick reaction time is needed, etc.). Flow parsing is a much broader process and addresses far more than moving object detection. I would suggest rephrasing some of these statements and being cautious with language like "advantages over" flow parsing because the proposed local mechanism is more specialized than flow parsing.

We do agree that the two mechanisms are synergistic and complementary, and hadn’t meant to imply that this mechanism could replace flow parsing (nor did we say that explicitly). Nevertheless, we see the reviewer’s point that some of our language might be construed in that direction. We have now modified the text in all of the places noted by the reviewer, as well as several others throughout the manuscript, to emphasize the complementarity of the two mechanisms.

3. I think the limitations from the public review should be referenced/discussed in the paper.

As noted in responses to other comments, we have made several revisions to address these points, including a greatly expanded section about flow parsing in the Discussion (pp. 17-18).

4. Warren and Rushton (2009) have demonstrated that humans parse object motion from self-motion to a similar extent regardless of whether the background motion surrounding a moving object is present in a monocular display. Niehorster and Li (2017) found a similar result for stereo optic flow. Simulations by Layton and Fajen (2016; 2020) and Layton and Niehorster (2019) support the hypothesis that a global mechanism plays a primary role. Together, these findings suggest that local mechanisms may play a much smaller role, at least for flow parsing. How do these findings relate to the proposed mechanism?

We agree that a key difference is that flow parsing involves a global mechanism, as described by Warren and Rushton (2009), and we have also demonstrated this finding in monkeys (Peltier et al. 2020). In contrast, the mechanism described in this study can be computed locally within each portion of the visual field. We now devote a paragraph of the discussion (p. 17) to this issue.

5. I found Supplementary Figure 1b very helpful to develop an intuition about the displays (as was the included video). I know figure space is at a premium, but if possible, including Supplementary Figure 1b in the main manuscript would likely help readers because the stimulus design is rather complex.

We appreciate the suggestion, as we considered it carefully, but we prefer to keep this figure in the supplement.

6. Aspects of the paragraph related to opposite at the bottom of p. 29 sounded repetitive after reading the previous parts of the paper and I thought they could be condensed/removed.

With the substantially expanded discussion of flow parsing, we think that this paragraph is worth retaining and is not redundant, as it makes distinct points regarding reference frames and multisensory contributions. We have trimmed it down some, however (now on p. 18).

7. I found the motion parallax light blue color difficult to see in Figure 2. I would appreciate it if the authors replaced it with a more salient color.

Thanks for the suggestion. We have made the cyan color darker so that it is more visible.

8 Supplementary Figure 2: Why is the proportion correct so much less than 50% on the monocular task? Given that the monocular task is very challenging and there are two possibilities, I would have expected the proportion correct to be roughly 50%.

We thank Reviewers #1 and #3 for questioning this. It turns out that these control data were from the version of the task with 4 objects, rather than the version with two objects. Thus, the chance level of performance (now indicated in the figure) is 25%. This accounts for most of the mystery regarding the monocular condition. The remaining variation around the chance level is presumably just somewhat random as the monkey was only exposed to the monocular condition in 16% of trials across 3 sessions. The figure and caption have been modified to clarify this issue (now Figure 1—figure supplement 2).

9. p. 7. I don't think I understand the following statement and would appreciate clarification: "…animals were translated along an axis in the fronto-parallel plane that was aligned with the preferred-null axis of the neuron under study." Does fronto-parallel plane mean rotation about the Z axis (axis toward the stimulus)?

No, the fronto-parallel plane does not imply any rotation about the Z-axis. The fronto-parallel plane is a pretty standard term for the vertical plane that includes the interaural axis, and which is parallel to the plane of the display screen. This is now stated on p. 24 for additional context. To align the stimulus axis with the preferred-null axis, the entire stimulus was rotated around the naso-occipital axis (which is what we presume the reviewer means by the Z-axis), as was stated in Methods previously. We have also added some clarification of this axis (p. 24).